# Arsenic contamination of Bangladesh aquifers exacerbated by clay layers

Ivan Mihajlov [1,2], M. Rajib H. Mozumder [1,3,7], Benjamín C. Bostick [3], Martin Stute[3,4], Brian J. Mailloux [4], Peter S. K. Knappett [5], Imtiaz Choudhury[6], Kazi Matin Ahmed[6], Peter Schlosser [1,3,8] & Alexander van Geen [3✉]

Confining clay layers typically protect groundwater aquifers against downward intrusion of contaminants. In the context of groundwater arsenic in Bangladesh, we challenge this notion here by showing that organic carbon drawn from a clay layer into a low-arsenic pre-Holocene (>12 kyr-old) aquifer promotes the reductive dissolution of iron oxides and the release of arsenic. The finding explains a steady rise in arsenic concentrations in a pre-Holocene aquifer below such a clay layer and the repeated failure of a structurally sound community well. Tritium measurements indicate that groundwater from the affected depth interval (40–50 m) was recharged >60 years ago. Deeper (55–65 m) groundwater in the same pre-Holocene aquifer was recharged only 10–50 years ago but is still low in arsenic. Proximity to a confining clay layer that expels organic carbon as an indirect response to groundwater pumping, rather than directly accelerated recharge, caused arsenic contamination of this pre-Holocene aquifer.

---

[1] Department of Earth and Environmental Sciences, Columbia University, New York, NY 10025, USA. [2] Geosyntec Consultants, Huntington Beach, CA 92648, USA. [3] Lamont-Doherty Earth Observatory of Columbia University, Palisades, NY 10964, USA. [4] Environmental Sciences, Barnard College, New York, NY 10025, USA. [5] Department of Geology & Geophysics, Texas A&M University, College Station, TX 77843, USA. [6] Department of Geology, University of Dhaka, Dhaka, Bangladesh. [7]Present address: Gradient, Cambridge, MA 02138, USA. [8]Present address: School of Sustainability, Arizona State University, Tempe, AZ 85281, USA. ✉email: avangeen@ldeo.columbia.edu

Most of the rural populations of Bangladesh and several neighboring countries obtain their drinking water from shallow tubewells that often do not meet the World Health Organization guideline of 10 µg/L arsenic (As). Chronic exposure to As above this level has been linked to increased infant and adult mortality, inhibited intellectual and motor function in children, and significantly reduced household earnings[1–4]. In an effort to reduce As exposure, government and non-governmental organizations in Bangladesh have installed several hundred thousand deep (>150 m) community wells that are often, although not always, low in As[5–11]. Impermeable clay layers capping these deep low-As aquifers were deposited before the onset of the current Holocene epoch ~12 kyr ago and are widely seen as protective because they inhibit the downward flow of overlying high-As groundwater[12,13]. The present study of a more accessible pre-Holocene aquifer in an intermediate (40–90 m) depth range challenges this notion by considering biogeochemical reactions initiated by clay layers that could trigger the release of As to underlying aquifers[14–17].

Microbially-mediated reduction of iron (Fe) oxides coupled to organic carbon oxidation is held responsible for the widespread release of As from Holocene sediments to groundwater in the Bengal basin[7,18–23]. A similar process can occur in pre-Holocene sands where it is made apparent by the conversion of orange Fe(III) to gray Fe(II) oxides in response to a supply of organic carbon[24–27]. The sources of this organic carbon could be immobile plant matter co-deposited with aquifer materials or mobile reactive dissolved organic carbon (DOC) advected by groundwater flow. The relative importance of these two pathways for As release to groundwater is still debated[7,18,21,28–34].

Our detailed study of a site in Bangladesh illustrates that organic carbon within clay layers, defined here to include both clay and silt and are also referred to as mud[35], is a third source of reactive carbon that can be mobilized by distant pumping and result in the contamination with As of a pre-Holocene aquifer. The new data show that reactive carbon released by clay layers can instead drive chemistry changes in aquifers[14] and trigger the release of As to underlying groundwater. The new results offer the most direct evidence yet of a new mechanism for groundwater contamination with As triggered by pumping, which was inferred from observations elsewhere in Bangladesh, the Mekong delta of Vietnam, and the Central Valley of California[15–17].

## Results

### Failures of a community well.
The study was motivated by a sudden increase in As concentrations from <10 to 60 µg/L in groundwater from a hand-pumped community well (CW12) in 2005[36], i.e., 18 months after installing a 3-m long screen in orange sands of a pre-Holocene aquifer at 60 m depth in a village 20 km east of Dhaka (Fig. 1a). Local drillers guided by the orange color of sands commonly install household wells in the 30–90 m depth range in the study area (Fig. 1a) and elsewhere in the Bangladesh basin[9,25,36–39]. Leaks of high-As groundwater[40] that could have caused the increase were not detected by pumping sections of the well with an inflatable packer[36]. The second installation of a community well screened within a few meters of the initial well confirmed that the orange sands are capped by a 10-m thick layer of clay at the site but this well also failed after producing low-As water for several months (the failed well was replaced with a deeper low-As well at 90-m depth soon after the problem was detected for the second time). The failure of two separate wells, manually pumped at modest flow rates, indicates that leakage of shallow groundwater along the well annulus is unlikely to have been the cause of the increase in As concentrations. The concern that such failures could instead be an early sign of broader contamination of pre-Holocene aquifers within the cone of depression in groundwater levels due to massive deep pumping for the municipal water supply of Dhaka[41,42] led to further study and the monitoring reported here.

### Drilling and monitoring of the impacted aquifer.
Drill cuttings collected in 2010–11 while installing four nests of monitoring wells within a radius of 100 m of the failed community well confirm the presence throughout the area (Site M) of a 6–13-m thick clay layer separating a shallow aquifer composed solely of gray sands from a deeper aquifer containing both orange and gray sands (Fig. 2c). Lithologs also show that the layer of orange sand tapped by the failed community wells is at least 9-m thick throughout the area and intercalated between gray sands above and below (Supplementary Fig. 1). Concentrations of As in the four monitoring wells installed in the orange sand were <10 µg/L (Fig. 3e), including the monitoring well location (M-Middle) installed within <10 m of the failed community well. Monitoring wells in the gray sands overlying the orange layer in the same aquifer initially contained concentrations of As ranging from 20 to 250 µg/L, whereas concentrations in the layer of gray sand below were <5 µg/L. Similarly, concentrations of dissolved Fe were >5 mg/L in gray intervals overlying the orange sand layer, <1 mg/L in most orange intervals, and generally low also in the gray sand below the orange interval (Fig. 3f). Between 2011 and 2018, concentrations of As at the central nest of monitoring wells (M-Middle) remained low at 61 and 64 m, but two shallower monitoring wells at 41 and 51-m depth at the same location show worrisome increases in As concentrations from 50 to 150 µg/L and from 250 to 400 µg/L, respectively, over the same period (Fig. 1b). Both of these monitoring wells were installed in gray sand below the thick clay layer separating the aquifer from the shallower Holocene aquifer, which is consistently elevated in As (Fig. 3e and Supplementary Fig. 1). Simple linear extrapolation of the time series suggests As concentrations could have started to rise above the 5 µg/L level typical of uncontaminated pre-Holocene aquifers around 2009 and 2003 at 41 and 51-m depth, respectively.

### Depositional history of impacted aquifer.
Radiocarbon ($^{14}$C) dating and other sediment characteristics document the depositional history of the aquifer tapped by the failed community well. Within several interspersed thin layers of clay in the sandy 37–39-m depth interval below the main clay layer, plant leaves and pieces of charcoal and wood were recovered. In all but one case, $^{14}$C ages of this material were within a few decades of the bulk organic carbon ages of the associated clay, indicating that bulk clay reflects the $^{14}$C content of the atmosphere and can be used without reservoir correction (Supplementary Table 1). The depth profile of radiocarbon ages indicates steady accumulation of 40 m of sediment from 17 to 5 $^{14}$C kyr ago (Fig. 2a). The data indicate that the upper portion of the pre-Holocene aquifer was deposited ≥12 $^{14}$C kyr ago (Supplementary Table 1; Fig. 2a), which in calendar years corresponds to ≥14 ka when correcting for changes in the $^{14}$C content of the atmosphere[43], and corresponds to a period when sea level was still below its current level. A radiocarbon age of 35–38 $^{14}$C kyr for the clay layer at the bottom of the aquifer at 79-m depth indicates that the period of rapid sediment accumulation at the Late Pleistocene/Holocene transition, paced by the rise in sea level, was preceded by slower accumulation or perhaps even erosion[35,44]. The data confirm that deeper pre-Holocene sands can be gray[45] and yet be associated with low-As concentrations in groundwater, as reported elsewhere in the Bengal basin[7].

In spite of differences in color, the drill cuttings indicate that the entire pre-Holocene aquifer in the 40–70-m depth range, composed today of both gray and orange sands, was probably deposited under

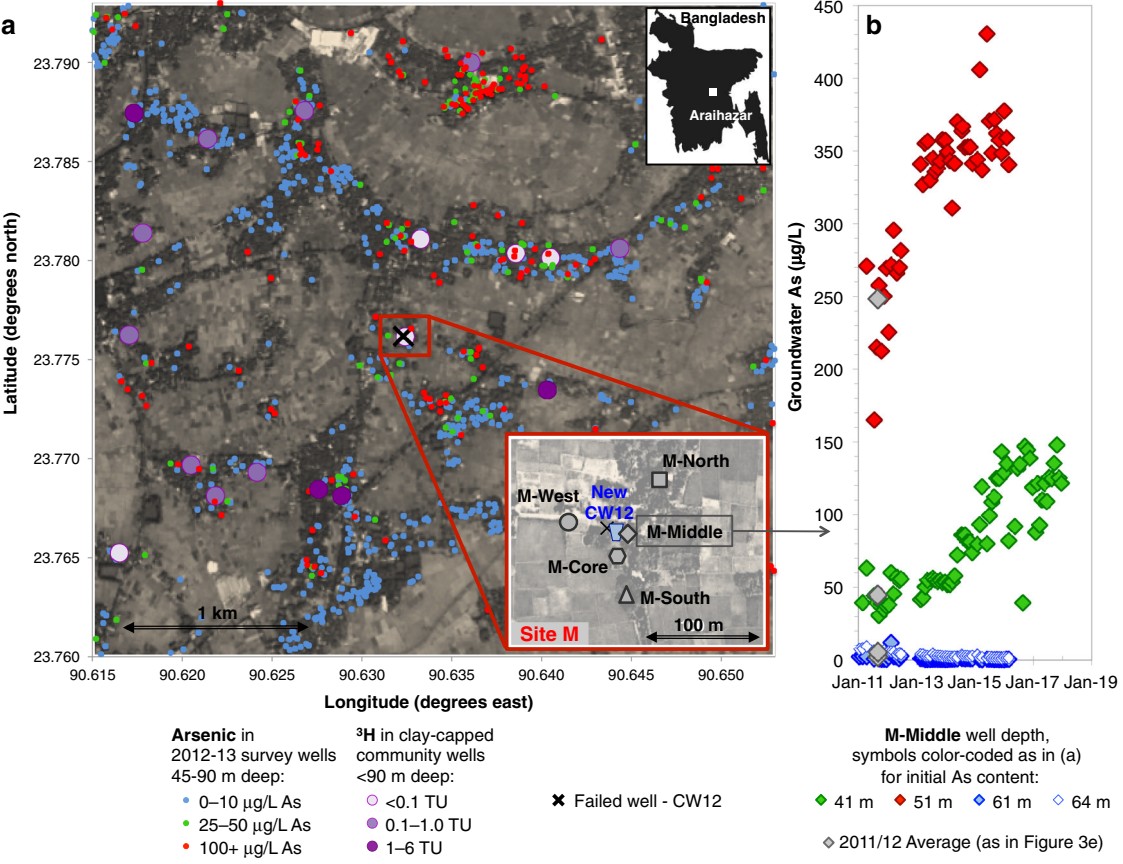

**Fig. 1 Regional and site map with tritium ($^3$H) and arsenic (As) distribution. a** Map of the field area and the focus site M. The small symbols denote the wells 45–90 m deep, surveyed in 2012–13 and color corded according to their As concentration measured using the ITS Arsenic Econo-Quick kit[37,70]. The large symbols denote the surrounding (2 km radius) community wells installed <90 m deep within a clay-capped low-As aquifer[36] and are color coded to reflect the highest measured $^3$H concentrations in tritium units (TU), as reported in Mihajlov et al.[48]. The enlarged inset map of site M displays sampled multi-level well nest locations, an additional coring location, and the location of community well (CW12) where arsenic concentrations rose twice prior to reinstallation at a greater depth. **b** Time-series of As concentrations in groundwater at the well cluster M-Middle from 2011 to 2017 (41 m) or from 2011 to 2016 (other depths); 2011/2012 average As concentrations plotted on Fig. 3e are also shown.

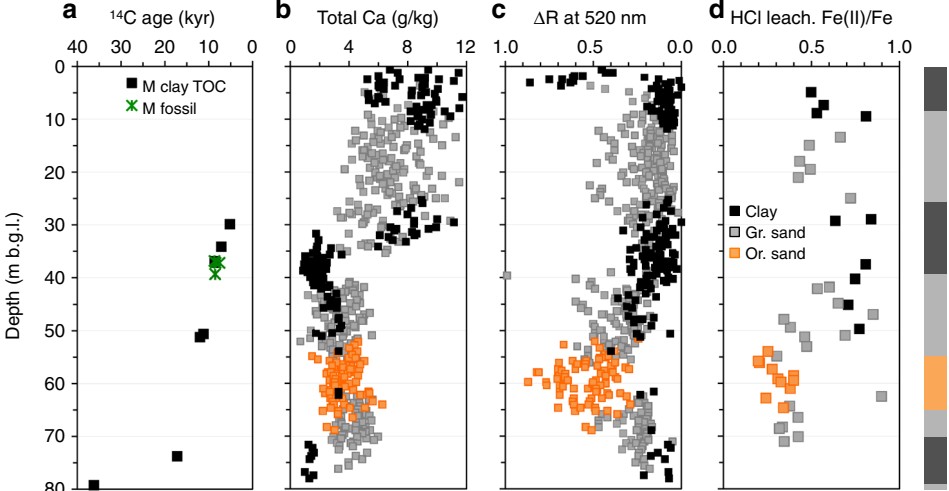

**Fig. 2 Site sediment vertical profiles. a** Conventional radiocarbon ($^{14}$C) ages expressed in thousands of years (kyr) measured on fossil plant material embedded in the sediment and/or total organic carbon of the clay layers. **b** Total calcium (Ca) content determined by X-ray fluorescence. **c** Diffuse spectral reflectance between 530 and 520 nm ($\Delta$R). **d** Percentage of Fe(II) within the total Fe extractable by 1 N hot HCl. Sand color, quantified by $\Delta$R and dictated by Fe speciation, is explicitly displayed to visualize orange and gray sand distribution. Results from the four multi-level well nest boreholes and the additional coring location are combined in the graphics and were used to prepare the generic site lithology displayed on the right.

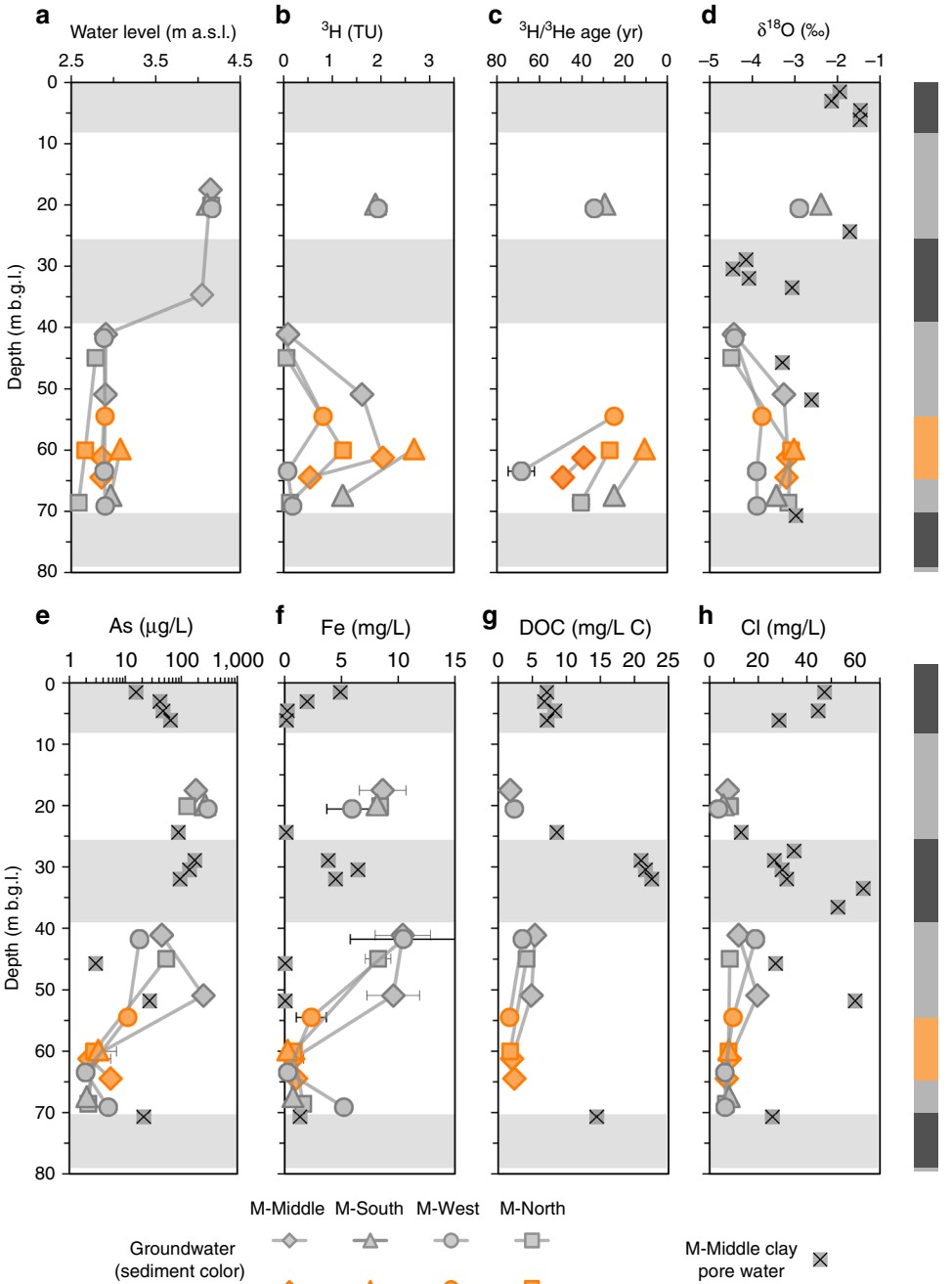

**Fig. 3 Vertical profiles of groundwater and clay pore water properties.** A generic site litholog is displayed on the right with shading in the panels indicating the extent of major clay/silt layers encountered. **a** Water levels are the annual average (December 2012–November 2013) groundwater elevations in meters above sea level. **b** Tritium ($^3H$) concentrations and **d** Oxygen-18 isotopic composition in water ($\delta^{18}O$) are one-time measurements with analytical error bars smaller than the symbol size. **c** Tritium-helium ($^3H/^3He$) ages were corrected for radiogenic He contribution and degassing at the time of sampling, where necessary; error bars indicate propagated analytical errors or standard deviations of the ages determined under different assumptions, whichever error is greater. **e** Arsenic, **f** iron, and **h** chloride concentrations in groundwater were averaged from discrete samples collected in 2011–2012 (arsenic data through 2016/17 from well nest M-Middle are shown in Fig. 1b); at depths where >3 samples were measured, standard deviations are also shown (As and Fe); Cl standard deviations are smaller than the symbol size. **g** Dissolved organic carbon concentrations.

similar conditions until 14 ka, when sea level was still quite low. There is no clear difference in grain size or exchangeable As content of gray and orange sands within the pre-Holocene layer (Supplementary Fig. 2). The calcium (Ca) content of cuttings from the entire pre-Holocene aquifer averages $3 \pm 2$ g/kg (1-sigma) between 40 and 70 m. The lower portion of the overlying clay layer to ~35-m depth contains even less Ca at ~1 g/kg (Fig. 2b). Above this interval, Ca concentrations sharply shift upwards and remain

elevated at $7 \pm 2$ g/kg throughout the shallow Holocene aquifer. This contrast has been observed elsewhere and attributed to the combination of authigenic precipitation of calcium carbonate in Holocene sands and extensive weathering of pre-Holocene sands while sea level was lower than today[26,46]. The key question is why concentrations of As in pre-Holocene sands below the clay layer have been rising since at least 2011 as this likely played a role in the repeated failure of the local community well.

**Contamination with arsenic by advection from shallow aquifer.** The shallow Holocene aquifer is one potential source of either As or organic carbon that could have triggered the local release of As through potential lateral discontinuities in the clay layer[47]. This pathway is a possibility given that, throughout the year, the water level in the aquifer tapped by the failed community well is at least 1 m below the water level in the shallow Holocene aquifer (Fig. 3a and Supplementary Fig. 4). The difference in hydraulic head is not driven by irrigation pumping, which only draws water from the shallow aquifer during winter months (Supplementary Fig. 4), but rather by massive pumping from the deep aquifer for the municipal water supply of Dhaka at a distance of 20–30 km to the west[41,42]. A steady intensification of this vertical hydraulic gradient over two decades has been documented at a site 2 km closer to Dhaka, where the difference is even larger than at the study site[48] (Supplementary Fig. 5).

Regional downward flow induced by massive pumping in Dhaka is confirmed by the penetration of the radioisotope tritium ($^3H$) produced by atmospheric testing of nuclear weapons in the 1950s and 1960s in portions of the pre-Holocene aquifer (Fig. 1a). Levels >1 tritium unit (TU) were detected within clay-capped orange sand in the 35–90 m depth range in four out of 18 community wells within a 2-km radius of the failed community well[36,48]. Concentrations of As were not elevated in these community wells, however, which is consistent with findings at the study site. Elevated levels of $^3H$ (>0.1 TU) were detected in a total of nine monitoring wells tapping the pre-Holocene aquifer in the 50–70-m depth range at Site M (Fig. 3b). The contribution of recent recharge is the largest in the orange sands at 50–60-m depth where groundwater As concentrations are low (i.e., in the middle of the pre-Holocene aquifer) and much lower to undetectable (≤0.1 TU) near the bottom of the clay layer capping the pre-Holocene aquifer.

One possible entry point for $^3H$-containing water to the 55–65-m depth range may be an area 500 m to the south of the study site where a thick clay layer capping the pre-Holocene aquifer is missing[49], but there may be other entry points. Profiles of groundwater ages based on the tritium-helium method show that the youngest ages of 10–40 years are focused in the orange sands at this site (Fig. 3c and Supplementary Note 1). Transport to these orange sands must be rapid since the $^3H$–$^3He$ ages bracket the age of groundwater in the shallow Holocene aquifer measured in the area (Fig. 3c)[50]. Adsorption to aquifer sediments has evidently been sufficient to delay any detectable influx of As or reactive DOC from the shallow aquifer to this portion of the pre-Holocene aquifer[26,46,51].

**Alternative mechanism for aquifer contamination with arsenic.** Contamination with As of the pre-Holocene aquifer is concentrated within a shallower and more reduced portion where there is little to no indication of recent recharge. The one exception is the well monitored at 51 m at nest M-Middle, which contains $^3H$ but, based on the observations from all the other wells, even the rise in As concentrations in this well is more likely to be driven by a process that is disconnected from recent recharge (Fig. 3b and Supplementary Note 1). If rapid advection from the shallow Holocene aquifer is not responsible, an alternative source of reactive DOC is required to explain the release of As within pre-Holocene sands. The pore water chemistry and tracers indicate that the thick clay layer could be this alternative source. In addition to the solid phase organic carbon content reaching 7% (Supplementary Fig. 2), the clay separating the Holocene and pre-Holocene aquifers contains pore water with DOC concentrations of up to 23 mg/L (Supplementary Note 2), i.e., one order of magnitude higher than in most of the

groundwater sampled by the monitoring wells (Fig. 3g). Generally unreactive tracers, such as chloride (Supplementary Note 3), sodium, and the stable isotopes of water (Supplementary Note 4), provide evidence of a flux of clay pore water across the interface separating the two units that is distinct from that of groundwater within the orange sands (Fig. 3d, h and Supplementary Fig. 3; Supplementary Table 2). Only in the case of chloride, however, was enough pore water extracted from the two clay intervals closest to the interface for analysis. The 20% contribution of clay water to the upper portion of the pre-Holocene aquifer estimated from chloride implies that advection of As contained in pore water from the clay alone cannot explain elevated levels and the rise of As concentrations in the monitoring wells.

## Discussion

Advection or diffusion out of the clays are two mechanisms through which DOC could be supplied from the clay to the underlying aquifer. In the case of advection, using a vertical difference of 1 m in hydraulic head across the 10-m thick clay layer (Fig. 3a) and a plausible range of vertical hydraulic conductivities for the clay of $10^{-9}$–$10^{-7}$ m/sec[39], the Darcy velocity of clay pore water into the pre-Holocene aquifer is on the order of 0.3–30 cm per year, i.e., 3–300 L/m$^2$ per year (Supplementary Note 5). This corresponds to a total organic carbon flux of 5–500 $\times 10^{-3}$ mol C/m$^2$ per year for a concentration of DOC in clay water of 20 mg/L (Fig. 3g). Assuming this flux magnitude over the past 20 years to reflect the trend in deep pumping and the development of the vertical hydraulic head difference (Supplementary Fig. 5), this corresponds to a total input of 0.1–10 mol C/m$^2$ into the upper portion of the pre-Holocene aquifer.

Fick's first law can be used to calculate the flux from diffusion. A much lower diffusive flux of DOC of 0.2–1 $\times 10^{-3}$ mol C/m$^2$ per year is calculated based on the difference in concentration of 17 mg/L spanning half of the thickness of the 10 m clay layer, the diffusivity constant of $9 \times 10^{-3}$ m$^2$/yr for acetate, and an effective porosity range of 0.1–0.5 (Supplementary Note 5). If the diffusion gradient was maintained over 5000 years by the continuous release of DOC from buried plant matter in the clay, the integrated flux of reactive carbon over this longer period corresponds to a total input of 1–5 mol C/m$^2$. The estimated flux of organic carbon estimated for advection over 20 years is therefore within the range of estimates for the diffusive flux over 5000 years.

Radiocarbon provides further evidence of the capping clay as a source of reactive DOC. The radiocarbon age of DOC of 4 kyr in the clay layer is comparable to that of DOC in the gray portion of the pre-Holocene aquifer and is unaffected by bomb radiocarbon input (Supplementary Fig. 3). In contrast, the DOC within the orange portion of the pre-Holocene aquifer is younger. Radiocarbon ages of DOC therefore point to the clay as the source but cannot differentiate between recent advection or long-term diffusion. Clay compaction linked to land subsidence caused by municipal pumping in Dhaka[52] could also be contributing by expelling reactive DOC. Such a mechanism has been invoked largely indirectly from broad-scale patterns to explain As contamination of groundwater in the Mekong Delta of Vietnam[15] and the Central Valley of California[17].

How does the magnitude of the DOC flux leaving the clay layer, by advection or diffusion, compare to what would be required to convert a 10–15-m thick layer of sand from orange to gray? This simple calculation constitutes an upper limit to what would be required to release As from pre-Holocene sands below the clay. If this layer never was as oxidized as the orange sand layer below it, then less DOC would be required. Sediment profiles from the site show that a shift in the acid-leachable Fe(II)/Fe ratio from 0.3 to 0.5 accompanies the change in sand color from

orange to gray (Fig. 2d)[19]. Given the average measured HCl-leachable Fe concentration of 5.6 g/kg for orange and gray sands in the pre-Holocene aquifer (about half the total Fe measured by X-ray fluorescence, Supplementary Fig. 2), ~40 moles of Fe would have to be reduced to change the color of 1 m³ of sand from orange to gray (Supplementary Note 5). Using a stoichiometric Fe/C ratio of 4 for reductive dissolution of Fe oxides[53], this means that an input of 1 mol C/m² would be able to change the color of only a ~0.1 m layer of aquifer sand. The flux of DOC, whether advected over the past 20 years or diffusing over 5000 years, is therefore insufficient by two orders of magnitude to reduce orange Fe oxides over the entire gray layer.

We offer two possible explanations for the apparent discrepancy. The first is that the 10-m thick upper layer of gray pre-Holocene sands below the clay layer may never have been oxidized completely, and thus would not have required as much reduction to release As into groundwater. This is a possibility because the relationship between the extent of reduction of Fe oxides in aquifer sands and As concentrations in groundwater is far from linear (Supplementary Note 6). The threshold of reduction associated with marked increase in groundwater As concentration is only reached when about a half of the sedimentary Fe oxides have already been reduced and the characteristic orange color of Fe(III) oxides has been lost[19,33]. The long-term diffusive flux of DOC could have contributed to approaching this threshold over several thousand years, with more recent Dhaka pumping providing the additional advective flux of DOC (Supplementary Note 7) to cross this threshold and cause the observed rise in groundwater As concentrations below the clay layer.

The second explanation relies on the observation that the bottom of the clay surface at the interface with the upper portion of the pre-Holocene aquifer varies in depth by as much as 5–10 m within 20–100 m of the failed community well after correcting for elevation differences at the surface (Supplementary Fig. 2). Combined with lateral flow, the diffusive input of reactive DOC from the bottom of a clay over time that varies in depth could have converted pre-Holocene sand from orange to gray in discrete intervals over a considerably wider depth range. We suggest this poised the aquifer for further reduction by DOC released from the clay layer and a rise in groundwater As concentrations around the time when local groundwater elevations started to show the impact of Dhaka pumping (Supplementary Fig. 5). Such additional reduction of even discrete intervals of the aquifer tapped by a long-screened well would be sufficient to contaminate with As the water drawn at the pump.

In summary, groundwater-As-concentrations rose over the past decade in a pre-Holocene aquifer capped by a clay layer. Using multiple lines of evidence, such a rise is attributed here for the first time to the reduction of Fe oxides driven by a flux of reactive carbon originating from a clay layer linked in turn to deep pumping at a considerable distance. The stoichiometry of Fe reduction by organic carbon suggests that the upper portion of the pre-Holocene aquifer either was fully oxidized and/or that DOC was released by neighboring clay layers over a wider depth range. In this particular area where the hydrogeology is clearly affected by Dhaka pumping, direct downward advection of As from the shallow aquifer is evidently not the cause of contamination of the pre-Holocene aquifer below the clay layer[26,46,51].

Our findings are of concern locally because many households within the Dhaka cone of depression are privately re-installing their wells to relatively shallow pre-Holocene aquifers[37]. Even in the absence of deep pumping, long-term diffusion of DOC from clay layers could explain why private wells screened just below a clay layer in other sedimentary aquifers are more likely to be contaminated with As than deeper wells with longer screens[54].

With groundwater pumping from sedimentary aquifers expected to continue throughout the world, more attention should be paid to potential contamination of groundwater with As by compacting clay layers[15,17].

## Methods

**Site description and installation.** The study site (23.7760°N and 90.6325°E, Fig. 1a), referred to throughout as site M, is located in Araihazar upazila, a subdistrict of Bangladesh located ~25 km east of the capital, Dhaka. The site is ~750 m southwest of the village Baylakandi/site B (Fig. 1a); both this region and site B, in particular, were described in detail by van Geen et al.[55], Zheng et al.[39], and Dhar et al.[56]. Four multi-level observation well nests were installed at site M in the winter of 2010/11, with 1.5 m long well screens strategically placed to monitor all major depth zones of the pre-Holocene aquifer (bottom of well reported as well depth). In the shallow aquifer, observation wells either had long screens permeating the entire aquifer (nests -Middle and -North, middle of screen reported as well depth), or 1.5 m long screens installed approximately mid-depth through the shallow aquifer (nests -South and -West, bottom of well reported as well depth). An additional location (M-Core) was drilled for the collection of sediment cuttings. The elevations of top of well casings relative to the reference well (shallow well at nest M-Middle: M-M.1) were determined visually within ±1 mm by leveling with a transparent, flexible U-tube filled with water. All well depths reported, thus, are relative to the M-M.1 top of casing. Well M-M.1 was, in turn, leveled by the same method to top of casing of well BayP7 at site B, for which the absolute elevation above sea level is known[39]. Thus, measured hydraulic head elevations could be referenced to the absolute elevation above sea level (m asl).

**Sampling and analyses of solid materials.** Sediment cuttings were collected at 0.6 m (2 ft) or 1.5 m (5 ft) intervals while drilling by the traditional hand-flapper or sludger method[19,57] to install the wells. This method biases samples slightly towards the coarser fraction, especially when sand and silt are mixed. Cuttings were described by grain size (clay, silty clay, or sand) and by sediment color (gray or orange) to construct lithol021s. On the day of collection, diffuse spectral reflectance between 530 and 520 nm was measured on the cuttings wrapped in Saran wrap to indicate the Fe speciation in the solid phase[19]. The cuttings were also analyzed by X-ray fluorescence (XRF) using a portable InnovX Delta instrument for total elemental concentrations of Ca and Fe contained within the sediment. Samples were run without drying or grinding to powder, and the internal calibration of the instrument was checked before and after each run by NIST reference materials SRM 2709, 2710, and 2711. A subset of ~26 and 21 cuttings from representative depths at well nests M-Middle and M-West, respectively, were additionally subjected to same-day extractions by a hot 10% (1.2 M) HCl leach for 30 min to release Fe from amorphous Fe minerals[19]. The acid leachates were analyzed immediately for Fe(II) and total Fe concentrations by ferrozine colorimetry[58]. Separate samples from the same set of cuttings were also subjected to a same-day extraction in a N₂-purged 1 M NaH₂PO₄ solution (pH~5) at room temperature for 24 h[59]. The phosphate extracts were analyzed for As by high-resolution inductively coupled plasma-mass spectrometry (HR ICP-MS).

While drilling through clay and silt layers, various leaf fragments, pieces of wood, a piece of charcoal, and select samples of clay itself (for bulk organic carbon) were preserved in zip-lock bags for ¹⁴C dating and ¹³C isotopic analysis. ¹⁴C/¹²C and ¹³C/¹²C analyses were performed at National Ocean Science Accelerator Mass-Spectrometer (NOSAMS) facility of Woods Hole Oceanographic Institution following standard protocols[60]. Radiocarbon data were reported as fraction modern (FM) ¹⁴C, with measurement errors listed in Supplementary Table 1. The values of ¹³C/¹²C were calculated as deviations in per mil (‰) from the Vienna Pee Dee Belemnite standard (δ¹³C$_{VPDB}$), with analytical errors typically <0.1‰. Radiocarbon ages were calculated using 5568 years as half-life of ¹⁴C[61] and no reservoir corrections or calibration to calendar years were made.

Clay samples on which ¹⁴C and ¹³C analyses of bulk organic C were performed, as well as 17 other representative sand and clay samples from various depths at site M, were refrigerated and analyzed ~2 years later for C content in the sediment. Total C (TC) and inorganic C (IC) in the sediment were measured on the solid analysis unit of a Shimadzu carbon analyzer, and the difference between the two measurements was reported at total organic C (TOC) percentage in the sediment. Quantification limits for TC were 0.06% and 0.03% (% of total sediment) in clay and sand samples, respectively, while the respective limits of IC analyses in clay and sand were 0.02% and 0.01%.

**Hydraulic head measurements.** Variations in hydraulic heads relative to the top of the well casing were manually measured on a monthly basis using a Solinst Model 101 meter. Monitoring in some wells started in January 2011, but monthly readings were taken in all M wells simultaneously starting in July 2011. The reported annual average water levels (Fig. 3a) include readings from December 2012 to November 2013. Submersed pressure transducers with data loggers (Levelogger, Solinst) were used to record long-term water levels and barometric pressure at 20-min intervals in select wells at M-Middle nest starting in February

2011. All water level elevations are reported in meters above sea level (m asl; see above for elevation leveling).

**Chemical measurements in the field**. Groundwater was sampled in April 2011 for pH, oxidation/reduction potential (ORP), temperature and conductivity in a tight flow-through chamber (MP 556 from YSI, Inc.) equipped with appropriate probes until the readings were stable. At the same time, dissolved oxygen was measured with a CHEMet kit and alkalinity samples were obtained by Gran titration[62]. Dissolved inorganic carbon (DIC) values were then calculated from the concurrently measured pH values and alkalinity. Ammonia was measured in select M nest wells using a $NH_3$ electrode (AmmoLyt$^{Plus}$ 700 IQ from YSI, Inc.) in May 2012.

**Clay pore water collection**. Clay pore water samples were collected in May 2012 by squeezing clay cuttings from a borehole drilled near well nest M-Middle. Immediately upon the clay cutting collection and the squeezing of 2–20 mL of pore water, the pore water samples were filtered through 0.45 μm syringe filters (Whatman 6753-2504) and processed for various analyses, described below, in the same way as groundwater samples.

**Incubation experiments and DOC and DIC analyses**. Dissolved organic carbon (DOC) samples were collected in May 2012, immediately filtered through the 0.45 μm syringe filters into glass vials, and acidified to 1% HCl. Some of the samples were purposefully left unacidified in tightly capped vials filled without a headspace of air, then analyzed for DOC one month later. The DOC that decayed in unacidified samples was calculated by subtracting DOC levels of unacidified samples from those of acidified samples and expressed as % reactive DOC. DOC (from all M samples) and DIC (clay pore water only, unacidified) were measured in triplicates on a Shimadzu carbon analyzer to a precision of <5% for most samples, and the average was reported.

**Sampling and groundwater analysis**. Groundwater samples for major and trace elemental analysis by HR ICP-MS were collected on a monthly basis from July 2011 to June or August 2012 from certain wells, for which a time-series average and standard deviation is reported; for other wells, 1–5 samples were collected over a period between April 2011 and December 2012, and their time-series average is reported without standard deviations, unless >3 samples were analyzed. Additionally, monthly samples for As analysis were collected from the wells at nest M-Middle from February 2013 to December 2017 (well screened at 41 m depth) or March 2016 (the remaining wells at the nest). All samples were acidified to 1% HCl in the laboratory at least one week prior to the analyses of Na, K, Ca, Mg, P, As, Fe, Mn, Sr, and Ba using HR ICP-MS[62,63] to a precision of <10% and accuracy of <10% when compared to internal laboratory reference standards. Groundwater samples for anion analysis were collected at the same time as the HR ICP-MS samples, but were not acidified, and only a subset of 1–8 samples per well were analyzed for the period of April 2011–July 2012. Anion samples were analyzed for Cl, $SO_4$, and F using ion chromatography, with a precision of <5% for Cl and 5–15% for $SO_4$ and F. The anion results are reported as averages of time series, with time-series standard deviations reported only where >3 monthly samples were analyzed.

**Stable isotopes (δ$^2$H and δ$^{18}$O) in water**. Samples for stable isotope ($^2$H and $^{18}$O) measurements were collected in 60 mL glass bottles with polyseal-lined caps in April 2011 (site M groundwater) and May 2012 (site M clay pore water). They were analyzed on a Picarro Isotopic Water Analyzer at Lamont-Doherty Earth Observatory with a precision of ±0.01–0.07‰ (δ$^{18}$O) and ±0.01–0.24‰ (δ$^2$H) (Supplementary Table 2). The values were reported in per mil (‰) differences from the Vienna Standard Mean Ocean Water values (VSMOW).

**Radiocarbon ($^{14}$C) and $^{13}$C of DIC and DOC**. Samples for the analysis of $^{14}$C and $^{13}$C were collected in 125 mL (DIC) or 250 mL (DOC) glass bottles with Polyseal-lined caps in April 2011 (site M DIC) and October 2012 (site M DOC). They were preserved with mercury chloride (DIC) or acid (1% HCl final concentration, DOC) to arrest potential biological processes after collection. The three clay pore water samples for $^{14}$C and $^{13}$C in DOC were much smaller (5–10 mL) and collected in May 2011. All radiocarbon and $^{13}$C analyses were performed, and the results reported (Supplementary Table 3), as described above for sediment samples.

**Tritium ($^3$H) and noble gas sampling**. The atmospheric testing of nuclear weapons released $^3$H, a radioactive isotope of H that peaked in the early 1960s, which made it possible to date groundwater recharged since the onset of tests by the $^3$H/$^3$He technique[64–66]. Samples for $^3$H/$^1$H measurements were collected in 125 mL glass bottles with polyseal-lined caps and analyzed at Lamont-Doherty Earth Observatory's Noble Gas Laboratory using the $^3$He ingrowth technique[67,68]. The analytical precision and detection limit of the $^3$H measurements were ±0.03–0.06 TU (Supplementary Table 4) and 0.05–0.10 TU, respectively, ($^3$H/$^1$H ratio of $10^{-18}$ corresponds to 1 TU). Samples for He and Ne isotopic measurements were collected in ~1 cm outer diameter soft copper tubes that hold ~19 cm$^3$ of groundwater. Concentrations of He, Ne, and $^3$He/$^4$He were measured by mass spectrometry[69] at Lamont-Doherty Earth Observatory's Noble Gas Laboratory,

with typical analytical precisions of ±0.05–0.10% for He and Ne concentrations and ±0.6–0.7% for $^3$He/$^4$He ratio.

## Data availability

Data that support the findings of this study that are not already included as tables in the paper will be deposited upon acceptance at https://www.hydroshare.org/

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

## Acknowledgements
Columbia University and the University of Dhaka's research in Araihazar, Bangladesh has been supported since 2000 by NIEHS Superfund Research Program grant P42 ES010349. NSF Coupled Natural and Human Systems Dynamics grant ICER 1414131 provided additional support. We thank M. S. Shahud, M. M. Hosain, and the villagers at site M for their help in the field, L. Baker, R. Friedrich, and R. Newton for data acquisition help, and Y. Zheng, H. Michael, and C. F. Harvey for their ideas and comments. This is Lamont-Doherty Earth Observatory contribution number 8396.

## Author contributions
I.M., B.C.B., M.S., and A.v.G. designed the study and conducted the fieldwork. B.J.M., P.S.K.K., M.R.H.M., and I.C. provided field and laboratory assistance. B.C.B., M.S., K.M.A., P.S., and A.v.G. advised and supported the work of I.M. I.M., and M.R.H.M analyzed the data and, with A.v.G., wrote the manuscript, which was then edited by all co-authors.

## Competing interests
The authors declare no competing interests.
