## [Peer Review File · Nature Communications]

Editorial Note: Parts of this peer review file have been redacted as indicated to remove third-party material where no permission to publish could be obtained.

Reviewers' comments:

Reviewer #1 (Remarks to the Author):

What are the major claims of the paper?

The authors make two claims, the first of which seems to be (both in the text and the title) the main focus of the paper:

[1] Withdrawing water from (particularly shallow) Pleistocene aquifers in Bangladesh (and similar environments) draws reactive organic carbon from clay layers within the aquifers promoting Fe(III) reduction and consequent mobilisation of arsenic. Thus, such extraction makes (particularly shallow) Pleistocene aquifers with nearby organic-rich clay layers prone to future increases in groundwater arsenic.

[2] Recharge induced by groundwater pumping may protect low-arsenic aquifers in Bangladesh (and similar environments)

Are they novel and will they be of interest to others in the community and the wider field? If the conclusions are not original, it would be helpful if you could provide relevant references.

[1] Claim [1] as expressed in the abstract is not novel - as the authors point out themselves - the idea having already been proposed by (Reference 15) Erban et al (2013) PNAS. www.pnas.org/cgi/doi/10.1073/pnas.1300503110 and (Reference 16) Planer-Friedrich, B. et al. (2012) Applied Geochemistry doi:<https://doi.org/10.1016/j.apgeochem.2012.08.005> amongst others, with the underlying importance of the process of reactive organics in aquitards driving microbially mediated reduction process in adjacent aquifers (albeit not with the same precise context) having been identified earlier by (Reference 14) McMahon and Chapelle (1991) Nature.

[2] To the best of my knowledge, claim [2] is novel with respect to the particular arsenic-releasing mechanism outlined in claim [1] - although the potentially arsenic-immobilising influence of relatively oxidising, perhaps SO₄ and/or NO₃ rich, surface waters has been widely known for many years - and - in contrast - the converse role of reactive organic carbon contained within such surface-derived waters has also long been muted as a driver of arsenic mobilisation (Harvey et al., 2002, Science).

Is the work convincing, and if not, what further evidence would be required to strengthen the conclusions?

The reported work supporting claim [1] is very convincing, supported in particular by compelling ¹⁴C and ³He/³H data as well as plausible back-of-the-envelope calculations as to the potential magnitude of the FeR and arsenic mobilisation impacts of the clay-later derived organic carbon that might be released.

Claim [2] is rather more speculative and the evidence presented for it in the manuscript is not, in my opinion, very strong given the likely 3-D heterogeneity of the studied aquifer system, the multiple biogeochemical processes involved and the reliance on a relatively restricted field area.

On a more subjective note, do you feel that the paper will influence thinking in the field?

As a generally very well written account of a carefully designed research programme aimed to explain secular changes in well water arsenic in a studied well, this is an interesting case study that supports the importance of reactive organic carbon in (clay) aquitards in controlling the mobilisation of arsenic in immediately adjacent aquifers.

Of particular importance is that the authors have highlighted that the main mechanism proposed is most applicable to SHALLOW Pleistocene aquifers - and that is a very useful practical message - whilst it may also be inferred from the manuscript that application to predicting secular changes in DEEPER Pleistocene aquifers would NOT result in robust predictions - indeed I would suggest that the potential dangers of doing this should be made clearer in the manuscript.

Further questions and concerns about the paper.

Much of the discussion is based upon the assumption that all the studied Pleistocene aquifers were "orange" not "grey" - this seems highly speculative and based on little evidence - the follow argument is not very convincing in that regard: "In spite of differences in color, the drill cuttings indicate that the entire Pleistocene aquifer in the 40-70 m depth range, composed today of both grey and orange sands, was probably deposited and subsequently flushed under similar conditions and therefore uniformly orange."

The use of the term "groundwater age" needs to be qualified - see for example the incisive commentary on this by Bethke and Johnson (2008) *Annu. Rev. Earth Planet. Sci.* 2008. 36:121–52

The observed change in well water arsenic seems to have occurred at most over a matter of years - this timescale seems faster than any of the advective or diffusive or other mechanisms proposed - although the arguments are otherwise highly plausible, this would seem to somewhat undermine the main claim of the manuscript and, in any event, this is a point that should be substantively addressed.

Appropriateness and validity of any statistical analysis

Reviewer #2 (Remarks to the Author):

Dear Editor, dear authors

The subject of the manuscript is the potential (near-)future contamination by As of groundwater from relatively shallow Pleistocene aquifers. The investigated aquifer, and the type of aquifer for which the conclusions are most relevant, are in specific aquifers which are typically within the shallow-most oxidized zone. These Pleistocene aquifers host the screens of an increasing number of private tube wells. Drillers have learned to install screens of drinking water wells in orange sands rather than in grey sands. In fact, they have learned to do so by other excellent works of authors from this same group.

The authors present data which indicate a risk of As contamination due to reductive dissolution of As-hosting iron oxides by DOC released from a low-/semipermeable sediment unit capping the studied shallow Pleistocene aquifer. They provide evidence that the risk is due to an increased downwards hydraulic gradient due to groundwater extraction for Dhaka's water supply.

Albeit from just one field site, the manuscript contains several data types, obtained from four well

nests installed in the area, for example: hydraulic head, arsenic conc. dynamics during about six-seven years, general major chemistry, stable isotopes of water and carbon, tritium and tritium/helium dating, ¹⁴C dating of sediments, DOC and DIC, wet chemical extractions of drill cuttings. Indeed, the conclusions are reasonably well documented by the data but still the presentation need to be improved and need to be more focused. For example, it should make a reader able to clearly identify evidence in the form of hard facts (e.g., isotopic data, etc.) from softer/discussion points (e.g. downwards DOC loading amount and timing, and whether the variable elevation of the clay-to-sand interface makes up for an otherwise small DOC loading).

While, of course, pointing out a potential risk is important, so – I believe – is it for nat. comm. that this risk is real and reasonably likely to occur (not necessarily a ‘ticking bomb’ but something in that direction though). My first of two major concerns is related to how well the ‘DOC transport’ step in the cause-and-effect chain (pumping>increased downward hyd. grad.>DOC transport>iron oxide reduction>As release) is adequately clearly proven. Importantly, this may be just a question of presentation though (See ‘major issue 3’ below).

Nevertheless, there’s currently some speculation (lines 227-230) regarding the effect of the variable elevation of the clay-to-sand interface in this area. If this is maintained in a revised version then I propose this should be backed up by some hydrological (vertical 2d?) transport modelling of dispersion should be applied, taking into account the horizontal groundwater movement due to Dhaka’s pumping. Which dispersivities are required in such a model to create the mixing depth intervals inferred by the data? Are these dispersivities reasonable? Without significant dispersion, flow lines in a confined aquifer would be quite nicely parallel to the upper and lower boundaries of the aquifer no matter its variation. The flow lines wouldn’t follow a specific elevation above sea level. But see my remaining comments – as I see it the authors could work around this issue in another way.

My second concern is related to the relevancy outside the specific field site. A previous study (Mihajlov et al., 2015, WRR) by authors of the same group was conducted with at least much of the same data types available and from a more extensive area (~25 km²) comprising the smaller study area of the current study. In Mihajlov et al., none of the samples from <90 m depth showed “evidence of As intrusion from the shallower depths”. I know that this is not the same as leakage of clay pore water and its DOC (and/or As?; see ‘Minor issue 3’). Yet I assume the authors would have made a careful statement in Mihajlov et al. if they sensed a risk of contamination by DOC which could challenge their statement. In Mihajlov et al., the few samples that had As concentration ≥10 µg/L were from gray-tinted aquifer sands (i.e., classified as not-orange) and apparently must have been classified as such at the time of well installation, as the samples apparently stem from actual drinking water wells. The conclusion in Mihajlov et al. therefore may appear to be in contrast to the conclusion of the present manuscripts. I acknowledge that there may be a difference between the data available to Mihajlov et al. back then (2015) versus now to the authors of the present manuscript. However, the authors should still clarify why conclusions regarding DOC leakage from clay units capping the <90 m Pleistocene aquifer could not be made by Mihajlov et al. (2015). Were the data required to make at least a minimum assessment of the risk not available to Mihajlov et al.? Perhaps this clarification can even be used to constrain the proposed risk, which would be a good thing.

If my above concerns can be dismissed by good arguments by the authors, and if the list of major and minor issues below can be dealt with in an appropriate manner, then I will strongly recommend that their work is published in Nat. Comm. In other words, formally I recommend the Editor to accept with ‘major revisions’.

Groundwater arsenic is more than deadly enough to justify publication (also in nat. comm.) even if it does not affect directly the 100,000s >150 m deep wells, but ‘only’ the numerous (how many?)

'shallow' low-As wells. For example, each of these shallower wells will need monitoring and probably the drilling of a deeper well in the future. These wells also are more likely private than the deeper wells, and this may limit the accessibility (legal) and economy required for proper monitoring, so new management systems need to be implemented.

I am sorry for my long review text – it could probably be boiled down to much much less, but I hope my points make it through anyways. I also would like to apologize beforehand for any misinterpretations made by me, as annoying they can be to have to defend then for the authors.

Finally, I sincerely hope and trust that the authors, a very strong group which I respect very much, will be able to adjust their manuscripts to make it an excellent and very important contribution to nat. comm.

Best regards
Søren Jessen

Major issues:

1. The manuscript 'sets the scene' nicely, but simplifies things too much by inferring direct relevancy to the 100,000s >150 m deep wells (lines 41-47, 121-122), while in fact the study addresses/collects data from a 'shallow' Pleistocene low-As aquifer. These are very important differences. The relevancy for the aquifers directly targeted should be made crystal clear from the beginning. I acknowledge that the summary (lines 274-276) do make this distinction but that is too late in my opinion. The notion in the abstract (lines 23-24) that by 'Pleistocene' the authors mean '>12 kyr-old' is too subtle. Some elaboration on this type of aquifer could be made. The studied oxidized aquifer may be Pleistocene in age by definition but at the same time it is younger than the last glacial maximum (LGM) during which the deepest Pleistocene aquifers likely became oxidized. The studied aquifer is therefore a 'post-LGM Pleistocene aquifer', and as such it is composed of much younger aquifer material, and its non-oxidized parts contain younger and more reactive organic carbon (and perhaps more mobile As) than the deeper and more flushed (pre-LGM) Pleistocene aquifers. The 14C sediment dates combined with BGS's old report's Fig. 3.4 indicates that perhaps "Hiatus 2" caused the oxidation of the studied aquifer sand during a few thousand years maximum (I'm no geologist, but... 'preboreal oscillation')? This affects for example lines 108-110 and elsewhere.

2. Is the carbon from the capping clay unit driving the As release from the confining unit? It most probably is, but the supporting data should be highlighted much more. For example, the similar 14C-age of DOC in the capping clay and the sand just below it vs. in the Holocene aquifer is not easily accessible to the reader. This is a hard fact, and should be presented as such, earlier in the text (than lines 204-208, repeated at 244-245, and 969-974). In general, all the hard facts (hydr. gradients, increase in As in 41 and 51 m depth, reactive DOC, water stable isotopes, other isotopic data, etc.) should be presented earlier and in the main text rather than in supporting info. Graphs of d-excess (= $\delta^{2H} - 8 \cdot \delta^{18O}$) vs. depth should replace 2H graphs. Interpretations of stable isotopes of water are usually made relative to the GMWL. Use of d-excess enables just that without the need for showing a δ^{2H} vs. δ^{18O} -plo. On the other hand the visual appearance of the 2H-vs.-depth plot does not directly add information because it will appear almost like the δ^{18O} -vs.-depth plot. Lowered (from ca. 10) or even negative d-excess results from evaporation.

3. Much of the manuscript discusses the possible transition of orange sands to grey in the upper part of the Pleistocene sand (lines 122-125 and elsewhere). The discussion is supported by reasonable but many and a bit discussion-prone back-of-the-envelope calculations. However, I don't see this redox state transition as necessary for the story: A transition to grey may be 'modern', or it may have

occurred as result of DOC diffusion during thousands of years, or the sand may have been deposited as oxidized and turned grey shortly after such as the Holocene deposits are now, so that no sea-level-linked transition ever occurred, or a combination of some of these. In any of these cases no one can take away the fact that right now arsenic concentrations are increasing, so it MUST be a modern phenomenon taking effect. The increasing As concentrations is the smoking gun of the story. Isotopic and other evidence further strongly indicate that leakage containing DOC from capping clay is the reason for this. Yes, the DOC becomes mobilized into a now grey (Pleist.) unit, but even so, and even if the unit is 'young Pleistocene', the unit's organic carbon is relatively old and unreactive due to flushing and the addition of more reactive org. carbon is totally likely able to mobilize arsenic (again?) from these sediments.

Along these lines, with the right argumentation neither the speculation in lines 227-230 nor the above proposed modeling may be necessary at all. This would help focus the paper.

Other options that would aid focusing the paper, and which the authors might consider are to (i) eliminate the Ca profile from Fig. 2 and the text dealing with it. Instead Fig. 2a could be increased in width; (ii) eliminate the color coding of the M-wells in Fig. 3. It shouldn't be important anyway, as they show the same trends (don't they?); (iii) perhaps aggregate data in depth-categorical plots, with categories like: 'Holocene reduced aqf.', 'Capping clay' (of Pleist. aquifer), 'Upper reduced Pleist. aqf.', 'Oxidized Pleist. aqf.', 'Lower reduced Pleist. aqf.'; 'Lower Pleist. clay'. If, for example, horizontal box plots were shown, statistical significance could be evaluated more easily by eye, or (iv) Fig. 3 could adapt the clever color coding of Fig. 2; (v) with focus only the capping clay and the Upper reduced and oxidized sand of the Pleistocene aquifer, all concentrations could be shown relative to the distance above/below the clay-aquifer-interface.

As all data goes into tables in the Extended Data, and as more detailed graphs can be provided also in ED where necessary, the simplification in the main figs. and text will only help communicate the main message.

4. The authors should present arguments that the case they present in the present manuscript is likely relevant across a much wider area, i.e. an area that extends even outside the 25 km² area of Mihajlov et al. (2015). This could (not necessarily 'should') include a figure/map, although some display of the distribution of Pleistocene 'shallow' low-As wells in Bangladesh delta indeed would be a major improvement to the manuscript, and increase its use in the future.

5. In the calculation of reactive DOC diffusion, the concentration gradient of the reactive DOC should be used; not the concentration gradient of total DOC (Lines 1022–1025 + 1113). Of course – one may then question the method by which reactive DOC was measured (one month reaction time within a filtered sample vs. some years in the subsurface with contact to solids and 'stuff') and from that dismiss this way to quantify reactive carbon.

Minor issues/questions:

1. Figs. ED2a (% Org. C) and ED7: Very important figures; shouldn't they be in the main text?

2. Line 115: Is there any grain size difference between pre-Holocene vs. Holocene aquifer sediments?

3. Lines 131-134, Fig. ED3d (PO₄ extractable As): (i) The main point with the figure I think would be to indicate that 'yes, mobilizable As is present in all depths – with the addition of reactive carbon, we may expect an As release'. With that as the only take home message, its displayed depth distribution shouldn't be given much (if any) attention. Now the figure seem to be include more or less just because it was measured. For example (ii) the symmetric feature of the depth-distribution is noted but actually not used in the interpretation or if it is, it is not clear how it is used. (iii) Also, if we chose to pay close attention to the actual values, then one could as well note that as much As can be mobilized from the capping clay as from the upper Pleist. grey sand. So, how (if) is it possible to distinguish

whether the As release take place in the Pleist. reduced aquifer (as proposed) or in the capping clay? Perhaps via the indicators of mixing (e.g., stable isotopes of water and Cl values)?

In my opinion, the distinction in (iii) above is not necessarily important. In both cases org. C from clay remains the driver for As release. Whether lowered hydraulic pressure induce leakage that carried DOC into the Pleist. aquifer or leakage that carried As released within the clay, is not pivotal.

4. Lines 135-136: (i) The origin and direction of reactive org. carbon could be from below as well as from above. Fig ED2a (% Org. C) ends right below the oxidized aquifer, so a source may exist deeper than the graph which we just don't see. The sentence somehow infer that we know the direction of groundwater flow during the last 20,000 years, which I don't think we do – it may have been upwards, depending on location of channels at the surface. (ii) With the text 'that turned orange sand to grey' it is inferred that a reduction of a deep oxidized (i.e., sea level related) sediments actually occurred. I don't think we can be sure that the sand did not just become grey very quickly after deposition, like the Holocene (and today's) sediment did (do) and was never oxidized after that. (iii) Delete last instance of 'that'.

5. Lines 240-242: Please elaborate, e.g. how important is this mechanisms thought to be in these other areas?

6. Lines 181-183: hard to evaluate for the reader. Statement seems a little too 'bombastic'. I suggest to soften it – or do the above mentioned box-plot option, which might reveal the clearest trend and ease its communication.

7. Line 169: Please clarify, 'is not responsible' for what?

8. Fig. ED7: Please change units to mM to become consistent with text.

9. Line 267: Please use 'Because' instead of 'As', not to confuse this 'As' with 'Arsenic'.

Reviewer #3 (Remarks to the Author):

This manuscript on possible in-aquifer exacerbation of arsenic release via clay layers in Bangladesh is thorough, well-written, well-designed and the implications are significant – particularly with regard to the protection, or otherwise, that clay layers above lower As Pleistocene aquifers may give. In general the manuscript seems worthy of publication in Nature Communications. Some points which would benefit from further consideration or clarification:

- It would be helpful to show more characteristics of what this "reactive carbon" source actually is. As far as I can tell, "reactive DOC" here is simply defined as a difference in DOC between unacidified for 1 month and immediately acidified subsamples (around line 941). Is this really a sufficient proxy for "reactive DOC"? This seems a rather big assumption, particularly as it is clear from other studies that not all types of DOC contribute to arsenic release comparably. How do these estimates of "reactive DOC" compare with other bioavailability proxies?

- The evidence presented strongly supports that contributions from the in-aquifer clay layer are contributing to arsenic release here. But do we know what exactly from that clay layer is driving this process? The contribution of potential methanogenesis is mentioned (lines 980 – 985) – is there any direct evidence to support this? (e.g. dissolved methane measurements?)

- The heterogeneity of the sub-surface clay layers seems of key importance here, as the authors acknowledge. If drill cuttings of 4 well nests within a 100 m radius indicate a 6 – 13 m thick clay layer, it seems plausible that this clay layer could be very thin, or even missing entirely, in nearby areas too (it is mentioned that the layer is missing 500 m south). Would it be possible that these fast tracks introduced by such windows in the clay layer could contribute surface-derived organics and modern recharge that contribute to the conditions seen here (e.g. tritium increases at 60 mbgl, mixing lines shown on Extended Data 5)? Can this be ruled out?

Reviewer #1 (Remarks to the Author):

What are the major claims of the paper?

The authors make two claims, the first of which seems to be (both in the text and the title) the main focus of the paper:

[1] Withdrawing water from (particularly shallow) Pleistocene aquifers in Bangladesh (and similar environments) draws reactive organic carbon from clay layers within the aquifers promoting Fe(III) reduction and consequent mobilisation of arsenic. Thus, such extraction makes (particularly shallow) Pleistocene aquifers with nearby organic-rich clay layers prone to future increases in groundwater arsenic.

[2] Recharge induced by groundwater pumping may protect low-arsenic aquifers in Bangladesh (and similar environments)

Are they novel and will they be of interest to others in the community and the wider field? If the conclusions are not original, it would be helpful if you could provide relevant references.

[1] Claim [1] as expressed in the abstract is not novel - as the authors point out themselves - the idea having already been proposed by (Reference 15) Erban et al (2013) PNAS. www.pnas.org/cgi/doi/10.1073/pnas.1300503110 and (Reference 16) Planer-Friedrich, B. et al. (2012) Applied Geochemistry doi:<https://doi.org/10.1016/j.apgeochem.2012.08.005> amongst others, with the underlying importance of the process of reactive organics in aquitards driving microbially mediated reduction process in adjacent aquifers (albeit not with the same precise context) having been identified earlier by (Reference 14) McMahon and Chapelle (1991) Nature.

McMahon and Chapelle (1991) and Planer-Friedrich et al. (2012) present in situ field data supporting the notion that reactive carbon can be released by clay, but only in the case of Planer-Friedrich et al. however in the context of As. Planer-Friedrich et al. propose a natural, monsoon-driven release of reactive carbon rather than a long-term worsening trend driven by pumping as we do in this paper. Some of our hydrogeology colleagues question this monsoonal mechanism but this is not a central issue for this paper and questioning would be a distraction. In contrast to McMahon or Planer-Friedrich, Erban et al. (2013) and Smith et al. (2018) infer a release of As quite indirectly - without the support of any direct measurements of organic carbon or the redox state in the clay or aquifer sands because no sediment cores were obtained as part of these studies. We believe that the combined evidence presented in our study (a well-documented, perturbed hydrogeological context as well as detailed geochemical profiles and time series) is therefore novel.

[2] To the best of my knowledge, claim [2] is novel with respect to the particular arsenic-releasing mechanism outlined in claim [1] - although the potentially arsenic-immobilising influence of relatively oxidising, perhaps SO₄ and/or NO₃ rich, surface waters has been widely known for many years - and - in contrast - the converse role of reactive organic carbon contained within such surface-derived waters has also long been muted as a driver of arsenic mobilisation (Harvey et al., 2002, Science).

The Harvey group at MIT has actually produced a number of papers (Neumann et al., Nature Geoscience 2010 - which is cited) arguing specifically that surface recharge, through the bottom of ponds in particular, supplies reactive carbon driving iron reduction, and therefore arsenic release, in shallow

Holocene aquifers. The issue has been highly debated by other groups in several subsequent studies. For this reason, we believe it is important to point out that, for whatever reason, rapid recharge of a Pleistocene aquifer was not associated with a detectable release of arsenic at our site.

Is the work convincing, and if not, what further evidence would be required to strengthen the conclusions?

The reported work supporting claim [1] is very convincing, supported in particular by compelling ^{14}C and $^3\text{He}/^3\text{H}$ data as well as plausible back-of-the-envelope calculations as to the potential magnitude of the FeR and arsenic mobilisation impacts of the clay-later derived organic carbon that might be released.

Claim [2] is rather more speculative and the evidence presented for it in the manuscript is not, in my opinion, very strong given the likely 3-D heterogeneity of the studied aquifer system, the multiple biogeochemical processes involved and the reliance on a relatively restricted field area.

We agree that the evidence of actual protection by recharge is less direct and have reworded the last sentence of the abstract accordingly:

“ Contrary to expectations, recharge accelerated by groundwater pumping is not associated with a release of arsenic, whereas proximity to a confining clay layer increases the risk of contamination.”

On a more subjective note, do you feel that the paper will influence thinking in the field?

As a generally very well written account of a carefully designed research programme aimed to explain secular changes in well water arsenic in a studied well, this is an interesting case study that supports the importance of reactive organic carbon in (clay) aquitards in controlling the mobilisation of arsenic in immediately adjacent aquifers.

Of particular importance is that the authors have highlighted that the main mechanism proposed is most applicable to SHALLOW Pleistocene aquifers - and that is a very useful practical message - whilst it may also be inferred from the manuscript that application to predicting secular changes in DEEPER Pleistocene aquifers would NOT result in robust predictions - indeed I would suggest that the potential dangers of doing this should be made clearer in the manuscript.

We focus on shallow Pleistocene aquifers in this study for two reasons: (1) aquifers in what we describe as the intermediate (40-90 m) depth range are increasingly tapped by households seeking low-arsenic water. In our broader study area of Araihasar upazila, for instance, there are 8,400 such intermediate wells (out of a total 49,000 wells) whereas there are little over 900 deep wells installed by the government (as shown in cited Jamil et al. 2019). (2) given limited access to mechanized drill rigs in Bangladesh, it is much easier (and less expensive) to collect clay and sand to ~90 m depth by modifying the local, manual drilling method. There is no particular reason to expect that the processes identified in this study do not apply to greater depth around major pumping areas such as Dhaka.

The first point was already stated in L. 269-273, in the last paragraph of the original submission, and the 2nd is now clarified in the first paragraph with the modified sentence:

“ The present study of a more accessible Pleistocene aquifer in an intermediate (40-90 m) depth range challenges this notion by showing that reactive carbon released by clay layers can instead drive chemistry changes in aquifers¹⁴ and trigger the release of As to underlying groundwater.”

Further questions and concerns about the paper.

Much of the discussion is based upon the assumption that all the studied Pleistocene aquifers were "orange" not "grey" - this seems highly speculative and based on little evidence - the following argument is not very convincing in that regard: "In spite of differences in color, the drill cuttings indicate that the entire Pleistocene aquifer in the 40-70 m depth range, composed today of both grey and orange sands, was probably deposited and subsequently flushed under similar conditions and therefore uniformly orange."

Without time travel, we agree that it is hard to prove with absolute certainty that the upper portion of the studied Pleistocene aquifer at this specific location was extensively flushed during the low Glacial low-stand in sea level and therefore at one point turned orange in color (the color of Fe(III) oxides, contrasting with grey mixed Fe(II/III) oxides. However, other sites drilled in the general area show Pleistocene orange sands located just below the same clay layer. We have now qualified the claim as follows, while pointing to additional evidence that the upper portion of the Pleistocene aquifer was well flushed by a hydraulic gradient accentuated by a still low sea level:

"The data indicate that the upper portion of the Pleistocene aquifer was deposited >12-14C kyr ago (Extended Data Table 1), which in calendar years corresponds to >14 ka, after correcting for changes in the 14C content of the atmosphere (ref.), and corresponds to a period when sea level was still well below its current level (Fig. 2a)."

"In spite of differences in color, the drill cuttings indicate that the entire Pleistocene aquifer in the 40-70 m depth range, composed today of both grey and orange sands, was probably deposited and subsequently flushed under similar conditions until 14 ka, when sea-level was still quite low, and therefore probably uniformly orange. Pleistocene sand still orange in color has been recorded at other drill sites in the area (Horneman et al., 2004)."

As correctly pointed out by Reviewer 2, knowing when the upper Pleistocene aquifer turned grey, or if it ever was orange, is not necessary to make the point that Dhaka pumping has at least accelerated the release of arsenic to groundwater because of the time series. The following sentence was therefore inserted in L. 245-247 of the revised version:

"Our interpretation therefore does not depend on whether the upper Pleistocene sands ever were truly orange or grey and reduced, but not to the point beyond which As is released to groundwater."

The use of the term "groundwater age" needs to be qualified - see for example the incisive commentary on this by Bethke and Johnson (2008) *Annu. Rev. Earth Planet. Sci.* 2008. 36:121–52

We rely primarily on bomb-produced ³H to distinguish portions of the Pleistocene aquifer before or after the 1960s. The calculation of 3H-3He ages is therefore not central to the argument. Extended Data Figure 5 explicitly makes the point that a water parcel is mixture of contributions of different ages, and specifies that the age reflects that of the younger, bomb-influenced component. We have added the Bethke and Johnson reference to the Supplementary Discussion.

The observed change in well water arsenic seems to have occurred at most over a matter of years - this timescale seems faster than any of the advective or diffusive or other mechanisms proposed - although

the arguments are otherwise highly plausible, this would seem to somewhat undermine the main claim of the manuscript and, in any event, this is a point that should be substantively addressed.

There is no reason to expect a linear trend in groundwater arsenic as iron oxides gradually become reduced. The available field data (e.g. cited Horneman et al., GCA 2004) suggest instead a threshold of reduction above which arsenic levels start to increase. As we already pointed out in l. 234-237 of the original submission, the upper portion of the Pleistocene aquifer could have turned grey gradually over several thousand years without a concomitant release in arsenic. What the time series clearly show is that Dhaka pumping caused the threshold of arsenic retention to be exceeded. There is therefore no inconsistency between the time scale and rate of change in arsenic concentrations with the diffusion/advection of reactive carbon triggered by Dhaka pumping.

Appropriateness and validity of any statistical analysis

Reviewer #2 (Remarks to the Author):

Dear Editor, dear authors

The subject of the manuscript is the potential (near-)future contamination by As of groundwater from relatively shallow Pleistocene aquifers. The investigated aquifer, and the type of aquifer for which the conclusions are most relevant, are in specific aquifers which are typically within the shallow-most oxidized zone. These Pleistocene aquifers host the screens of an increasing number of private tube wells. Drillers have learned to install screens of drinking water wells in orange sands rather than in grey sands. In fact, they have learned to do so by other excellent works of authors from this same group.

The authors present data which indicate a risk of As contamination due to reductive dissolution of As-hosting iron oxides by DOC released from a low-/semipermeable sediment unit capping the studied shallow Pleistocene aquifer. They provide evidence that the risk is due to an increased downwards hydraulic gradient due to groundwater extraction for Dhaka's water supply.

Albeit from just one field site, the manuscript contains several data types, obtained from four well nests installed in the area, for example: hydraulic head, arsenic conc. dynamics during about six-seven years, general major chemistry, stable isotopes of water and carbon, tritium and tritium/helium dating, ¹⁴C dating of sediments, DOC and DIC, wet chemical extractions of drill cuttings. Indeed, the conclusions are reasonably well documented by the data but still the presentation need to be improved and need to be more focused. For example, it should make a reader able to clearly identify evidence in the form of hard facts (e.g., isotopic data, etc.) from softer/discussion points (e.g. downwards DOC loading amount and timing, and whether the variable elevation of the clay-to-sand interface makes up for an otherwise small DOC loading).

The main site surrounding the repeatedly failed community well was studied in detail by drilling and installing four separate nests of wells. The site was also selected within an area with a large number of wells tapping the same aquifer (Figure 1a), a minor but significant portion of which are low in arsenic. Similar areas throughout Bangladesh where households were tapping the intermediate aquifer are identified in a map published by (cited) Jamil et al. 2019. We therefore feel the observations are relevant

to a much broader area affected by municipal pumping in Dhaka (and eventually probably other growing cities), in addition to the underlying processes being generalizable to other settings.

While, of course, pointing out a potential risk is important, so – I believe – is it for nat. comm. that this risk is real and reasonably likely to occur (not necessarily a ‘ticking bomb’ but something in that direction though). My first of two major concerns is related to how well the ‘DOC transport’ step in the cause-and-effect chain (pumping>increased downward hyd. grad.>DOC transport>iron oxide reduction>As release) is adequately clearly proven. Importantly, this may be just a question of presentation though (See ‘major issue 3’ below).

Nevertheless, there’s currently some speculation (lines 227-230) regarding the effect of the variable elevation of the clay-to-sand interface in this area. If this is maintained in a revised version then I propose this should be backed up by some hydrological (vertical 2d?) transport modelling of dispersion should be applied, taking into account the horizontal groundwater movement due to Dhaka’s pumping. Which dispersivities are required in such a model to create the mixing depth intervals inferred by the data? Are these dispersivities reasonable? Without significant dispersion, flow lines in a confined aquifer would be quite nicely parallel to the upper and lower boundaries of the aquifer no matter its variation. The flow lines wouldn’t follow a specific elevation above sea level. But see my remaining comments – as I see it the authors could work around this issue in another way.

We do not believe a detailed groundwater flow model is needed given the documented range of depths of the bottom of the clay layer in this area (Extended Data Fig. 1). Even if the flow lines in the confined aquifer are mostly horizontal, at some point they begin at the clay layer of variable depth and then continue horizontally. Rather than dispersion, we believe it is lateral advection that causes the “smearing” over the depth range of reactive DOC release from the bottom of clay layers. Due to the variable depth of the bottom of the clay, a portion of the lateral flow lines within the impacted upper (and grey) portion of the aquifer are expected to originate at the clay layer interface. The ^3H peak within the orange sand layer, without ^3H above or below, provides clear evidence of lateral transport in the area.

My second concern is related to the relevancy outside the specific field site. A previous study (Mihajlov et al., 2015, WRR) by authors of the same group was conducted with at least much of the same data types available and from a more extensive area (~25 km²) comprising the smaller study area of the current study. In Mihajlov et al., none of the samples from <90 m depth showed “evidence of As intrusion from the shallower depths”. I know that this is not the same as leakage of clay pore water and its DOC (and/or As?; see ‘Minor issue 3’). Yet I assume the authors would have made a careful statement in Mihajlov et al. if they sensed a risk of contamination by DOC which could challenge their statement. In Mihajlov et al., the few samples that had As concentration ≥ 10 $\mu\text{g/L}$ were from gray-tinted aquifer sands (i.e., classified as not-orange) and apparently must have been classified as such at the time of well installation, as the samples apparently stem from actual drinking water wells. The conclusion in Mihajlov et al. therefore may appear to be in contrast to the conclusion of the present manuscripts. I acknowledge that there may be a difference between the data available to Mihajlov et al. back then (2015) versus now to the authors of the present manuscript. However, the authors should still clarify why conclusions regarding DOC leakage from clay units capping the <90 m Pleistocene aquifer could not be made by Mihajlov et al. (2015). Were the data required to make at least a minimum assessment of

the risk not available to Mihajlov et al.? Perhaps this clarification can even be used to constrain the proposed risk, which would be a good thing.

As the reviewer points out, “evidence of As intrusion from the shallower depths” is not the same mechanism at all as leakage of DOC from clay layers. Nothing in Mihajlov et al. (2015) is contradicted by the findings reported in this new paper and the previous paper focused on drinking-water wells, most of them considerably deeper. Much of the drilling data presented in the present manuscript were included in the 2013 PhD thesis of I. Mihajlov, but the increase in groundwater As in a monitoring well depth at this site (Fig. 1) had not yet been detected at the time, or when the 2015 paper was published.

If my above concerns can be dismissed by good arguments by the authors, and if the list of major and minor issues below can be dealt with in an appropriate manner, then I will strongly recommend that their work is published in Nat. Comm. In other words, formally I recommend the Editor to accept with ‘major revisions’.

Groundwater arsenic is more than deadly enough to justify publication (also in nat. comm.) even if it does not affect directly the 100,000s >150 m deep wells, but ‘only’ the numerous (how many?) ‘shallow’ low-As wells. For example, each of these shallower wells will need monitoring and probably the drilling of a deeper well in the future. These wells also are more likely private than the deeper wells, and this may limit the accessibility (legal) and economy required for proper monitoring, so new management systems need to be implemented.

I am sorry for my long review text – it could probably be boiled down to much much less, but I hope my points make it through anyways. I also would like to apologize beforehand for any misinterpretations made by me, as annoying they can be to have to defend then for the authors.

Finally, I sincerely hope and trust that the authors, a very strong group which I respect very much, will be able to adjust their manuscripts to make it an excellent and very important contribution to nat. comm.

Best regards

Søren Jessen

Major issues:

1. The manuscript ‘sets the scene’ nicely, but simplifies things too much by inferring direct relevancy to the 100,000s >150 m deep wells (lines 41-47, 121-122), while in fact the study addresses/collects data from a ‘shallow’ Pleistocene low-As aquifer. These are very important differences. The relevancy for the aquifers directly targeted should be made crystal clear from the beginning. I acknowledge that the summary (lines 274-276) do make this distinction but that is too late in my opinion. The notion in the abstract (lines 23-24) that by ‘Pleistocene’ the authors mean ‘>12 kyr-old’ is too subtle.

We see no fundamental reason why the processes studied here in relatively accessible intermediate Pleistocene aquifers in the 40-90 m depth range should not be relevant to deeper wells, including the several hundred thousand wells installed by the government. The key requirement for relevance instead is deep pumping, which currently is limited primarily to metropolitan Dhaka (this may change in the future). In fact, we have new unpublished evidence that the deep aquifer is becoming contaminated already in a couple of villages in our study area where the vertical gradient is particularly large because

of a very thick clay layer capping the deep aquifer. We have clarified this in the abstract and introduction by inserting the following words shown (here) in **bold**:

*Abstract: “Confining clay layers are widely perceived to provide protection to **deep** Pleistocene (>12 kyr-old) low-arsenic aquifers against intrusion of shallower high-arsenic groundwater.”*

*“In an effort to reduce As exposure, government and non-governmental organizations in Bangladesh have installed several hundred thousand deep (>150 m) community wells that are often, although not always, low in As 5-11. Impermeable clay layers capping these **deep** low-As aquifers are widely seen as protective because they inhibit the downward flow of overlying high-As groundwater 12,13. The present study of **a more accessible Pleistocene aquifer in an intermediate 40-90 m depth range** challenges this notion by showing that reactive carbon released by clay layers...”*

Some elaboration on this type of aquifer could be made. The studied oxidized aquifer may be Pleistocene in age by definition but at the same time it is younger than the last glacial maximum (LGM) during which the deepest Pleistocene aquifers likely became oxidized. The studied aquifer is therefore a ‘post-LGM Pleistocene aquifer’, and as such it is composed of much younger aquifer material, and its non-oxidized parts contain younger and more reactive organic carbon (and perhaps more mobile As) than the deeper and more flushed (pre-LGM) Pleistocene aquifers. The ¹⁴C sediment dates combined with BGS’s old report’s Fig. 3.4 indicates that perhaps “Hiatus 2” caused the oxidation of the studied aquifer sand during a few thousand years maximum (I’m no geologist, but... ‘preboreal oscillation’)? This affects for example lines 108-110 and elsewhere.

Yes - the upper portion of the Pleistocene aquifer is post-LGM but not the lower portion. See similar comment from Reviewer 1 above, including the text modified in response.

2. Is the carbon from the capping clay unit driving the As release from the confining unit? It most probably is, but the supporting data should be highlighted much more. For example, the similar ¹⁴C-age of DOC in the capping clay and the sand just below it vs. in the Holocene aquifer is not easily accessible to the reader. This is a hard fact, and should be presented as such, earlier in the text (than lines 204-208, repeated at 244-245, and 969-974). In general, all the hard facts (hydr. gradients, increase in As in 41 and 51 m depth, reactive DOC, water stable isotopes, other isotopic data, etc.) should be presented earlier and in the main text rather than in supporting info.

Hydraulic heads, aquifer As, DOC concentrations, one of the water isotopes ($\delta^{18}O$) are already shown in main Figure 3. The comparable age of DOC within the clay and below is a useful constraint but, we believe, not sufficiently strong evidence to add yet another panel to Fig. 3.

Graphs of d-excess (= $\delta^2H - 8 * \delta^{18}O$) vs. depth should replace ²H graphs. Interpretations of stable isotopes of water are usually made relative to the GMWL. Use of d-excess enables just that without the need for showing a δ^2H vs. $\delta^{18}O$ -plo. On the other hand the visual appearance of the ²H-vs.-depth plot does not directly add information because it will appear almost like the $\delta^{18}O$ -vs.-depth plot. Lowered (from ca. 10) or even negative d-excess results from evaporation.

This is correct. We have therefore substituted δ^2H profiles with the profiles of excess deuterium in Extended Figure 3.

3. Much of the manuscript discusses the possible transition of orange sands to grey in the upper part of the Pleistocene sand (lines 122-125 and elsewhere). The discussion is supported by reasonable but many and a bit discussion-prone back-of-the-envelope calculations. However, I don't see this redox state transition as necessary for the story: A transition to grey may be 'modern', or it may have occurred as a result of DOC diffusion during thousands of years, or the sand may have been deposited as oxidized and turned grey shortly after such as the Holocene deposits are now, so that no sea-level-linked transition ever occurred, or a combination of some of these. In any of these cases no one can take away the fact that right now arsenic concentrations are increasing, so it MUST be a modern phenomenon taking effect. The increasing As concentrations is the smoking gun of the story. Isotopic and other evidence further strongly indicate that leakage containing DOC from capping clay is the reason for this. Yes, the DOC becomes mobilized into a now grey (Pleist.) unit, but even so, and even if the unit is 'young Pleistocene', the unit's organic carbon is relatively old and unreactive due to flushing and the addition of more reactive org. carbon is totally likely able to mobilize arsenic (again?) from these sediments.

Along these lines, with the right argumentation neither the speculation in lines 227-230 nor the above proposed modeling may be necessary at all. This would help focus the paper.

We responded above to this valid point, also raised by Reviewer 1.

Other options that would aid focusing the paper, and which the authors might consider are to (i) eliminate the Ca profile from Fig. 2 and the text dealing with it. Instead Fig. 2a could be increased in width; (ii) eliminate the color coding of the M-wells in Fig. 3. It shouldn't be important anyway, as they show the same trends (don't they?); (iii) perhaps aggregate data in depth-categorical plots, with categories like: 'Holocene reduced aqf.', 'Capping clay' (of Pleist. aquifer), 'Upper reduced Pleist. aqf.', 'Oxidized Pleist. aqf.', 'Lower reduced Pleist. aqf.'; 'Lower Pleist. clay'. If, for example, horizontal box plots were shown, statistical significance could be evaluated more easily by eye, or (iv) Fig. 3 could adapt the clever color coding of Fig. 2; (v) with focus only the capping clay and the Upper reduced and oxidized sand of the Pleistocene aquifer, all concentrations could be shown relative to the distance above/below the clay-aquifer-interface.

As all data goes into tables in the Extended Data, and as more detailed graphs can be provided also in ED where necessary, the simplification in the main figs. and text will only help communicate the main message.

We actually believe the Ca profile is a key piece of information that, along with the radiocarbon dating, allows us to claim that the upper and lower portion of the intermediate Pleistocene aquifer deposited and flushed under similar conditions (see also response to Reviewer 1 on this issue). However, we do agree with the reviewer that the color-coding was not particularly informative and perhaps even confusing. We have substituted by keeping the symbol shapes as a way of identifying the different drilling sites and nests of wells and coloring them according to the presence of grey or orange sand at the depth of the well screen. We have tried plotting properties as a function of depth below the clay but it complicates things quite a bit and risks disconnecting the reader from the field observations - which we believe are the strength of this contribution.

4. The authors should present arguments that the case they present in the present manuscript is likely relevant across a much wider area, i.e. an area that extends even outside the 25 km² area of Mihajlov et al. (2015). This could (not necessarily 'should') include a figure/map, although some display of the

distribution of Pleistocene ‘shallow’ low-As wells in Bangladesh delta indeed would be a major improvement to the manuscript, and increase its use in the future.

The area covered by Mihajlov et al. (2016) is actually considerably larger than 25 km². We make the point early on that “Local drillers guided by the orange color of sands commonly install household wells in the 30-90 m depth range in the study area (Fig. 1a) and elsewhere in the Bangladesh basin 9,25,36-39.”

In response to the comment, we modified the first sentence of the last paragraph so it refers to vulnerable aquifers throughout the Bengal basin, referring to our recent publication that maps these:

“ Practical implications of the mechanism of As release to groundwater newly documented here include the need for more frequent monitoring of the growing number of private wells installed in relatively shallow Pleistocene aquifers throughout Bangladesh ³⁷, especially where head gradients have been perturbed by deep pumping.”

5. In the calculation of reactive DOC diffusion, the concentration gradient of the reactive DOC should be used; not the concentration gradient of total DOC (Lines 1022–1025 + 1113). Of course – one may then question the method by which reactive DOC was measured (one month reaction time within a filtered sample vs. some years in the subsurface with contact to solids and ‘stuff’) and from that dismiss this way to quantify reactive carbon.

Even though the approach to estimating reactive carbon is similar to the procedure followed by the frequently quoted study of Neumann et al. (2010), it should not be taken too literally because such incubations do not replicate the anaerobic conditions of these aquifers. Given the uncertainties involved in other aspects of the calculation, we also do not think the distinction matters all that much.

Minor issues/questions:

1. Figs. ED2a (% Org. C) and ED7: Very important figures; shouldn’t they be in the main text?

We do not think the profile of solid phase organic carbon (ED Fig. 2) and the proportion of reactive DOC lost over time are central enough to crowd further the main figures (or replace any of the current panels).

2. Line 115: Is there any grain size difference between pre-Holocene vs. Holocene aquifer sediments?

There was no obvious difference, but it is worth remembering that the cuttings were obtained after washing away the fines to avoid contamination across intervals when using the local drilling method. There is therefore a selection towards coarser particles. In any case, we don’t think any subtle difference between Holocene and Pleistocene grain-size would be relevant to the story.

3. Lines 131-134, Fig. ED3d (PO₄ extractable As): (i) The main point with the figure I think would be to indicate that ‘yes, mobilizable As is present in all depths – with the addition of reactive carbon, we may expect an As release’. With that as the only take home message, its displayed depth distribution shouldn’t be given much (if any) attention. Now the figure seem to be include more or less just because it was measured. For example (ii) the symmetric feature of the depth-distribution is noted but actually not used in the interpretation or if it is, it is not clear how it is used. (iii) Also, if we chose to pay close attention to the actual values, then one could as well note that as much As can be mobilized from the

capping clay as from the upper Pleist. grey sand. So, how (if) is it possible to distinguish whether the As release take place in the Pleist. reduced aquifer (as proposed) or in the capping clay? Perhaps via the indicators of mixing (e.g., stable isotopes of water and Cl values)?

In my opinion, the distinction in (iii) above is not necessarily important. In both cases org. C from clay remains the driver for As release. Whether lowered hydraulic pressure induce leakage that carried DOC into the Pleist. aquifer or leakage that carried As released within the clay, is not pivotal.

As the original version stated “Moreover, the distribution of PO₄-extractable As on the grey sands both above and below the orange sand layer is symmetric, generally ranging from 0.3 to 1.0 mg/kg, which indicates that similar amounts of solid-phase As adsorbed on Fe oxides are available for mobilization both above and below the orange sand (Extended Data Fig. 2).” We think this is an important clue: if the extractable As concentration in the grey sand had been higher than in the orange sand (as in the grey Holocene sand above the clay for instance), it would have been harder to rule out that the solid As concentration did not play a role in explaining the difference in dissolved As content between the two portions (upper grey sand and middle orange sand) of the Pleistocene aquifer. The relatively low extractable As throughout the Pleistocene aquifer also supports the notion of a similar depositional and flushing history.

Profiles of As in the clay and underlying aquifer, combined with the conservative tracer Cl, suggest some of the increase in aquifer As could indeed be the result of the diffusion/advection of As (Fig. 3e/h and Extended Data Fig. 7). This does not apply to the conversion of orange sand to grey sand, however, because it requires an advected/diffusing electron donor that then drives the local reduction. This then probably released more As or at least prevented the readsorption of diffusing/advected As from the clay. We cannot distinguish the two scenarios based on the available data and, as the reviewer pointed out, it does not really matter in terms of the paper’s main point: the clay has released reactive carbon (possibly some As), the As remains in the water because the sand was reduced below the clay, which in turn released additional As.

4. Lines 135-136: (i) The origin and direction of reactive org. carbon could be from below as well as from above. Fig ED2a (% Org. C) ends right below the oxidized aquifer, so a source may exist deeper than the graph which we just don’t see. The sentence somehow infer that we know the direction of groundwater flow during the last 20,000 years, which I don’t think we do – it may have been upwards, depending on location of channels at the surface. (ii) With the text ‘that turned orange sand to grey’ it is inferred that a reduction of a deep oxidized (i.e., sea level related) sediments actually occurred. I don’t think we can be sure that the sand did not just become grey very quickly after deposition, like the Holocene (and today’s) sediment did (do) and was never oxidized after that. (iii) Delete last instance of ‘that’.

The bottom portion of the solid phase profile includes two samples from the underlying clay - which are quite low in % organic C. Given in addition the presence of a clay layer above and below the intermediate aquifer at this site, it is hard to envision a supply of reactive DOC from below. DOC could well be advected laterally, however, and in fact we invoke this mechanism to explain the thickness of the grey sand layer. We explained above why we believe the upper Pleistocene aquifer was flushed and orange initially.

We replaced “that” with “and”: “A shallow source of reactive carbon that turned orange sand to grey and is located only above the still orange sands is therefore more likely.”

5. Lines 240-242: Please elaborate, e.g. how important is this mechanisms thought to be in these other areas?

This is hard to tell. Neither the Mekong Delta or Central Valley included any study of core material. This is a distinguishing feature of our contribution.

6. Lines 181-183: hard to evaluate for the reader. Statement seems a little too ‘bombastic’. I suggest to soften it – or do the above mentioned box-plot option, which might reveal the clearest trend and ease its communication.

Perhaps the reviewer meant “forceful” or “assertive” or something similar? We feel that the supplementary section entitled “Estimate of the Contribution of Clay Pore Water to Upper Pleistocene Aquifer Using Chloride as a Tracer” justifies this qualified statement. Still, we qualified the statement as follows:

“Chloride concentrations in the clay pore water and the grey sands underneath (Supplementary Discussion) suggest that clay pore water accounts for approximately one third of the composition of groundwater just below the clay layer.”

7. Line 169: Please clarify, ‘is not responsible’ for what?

This was clarified as follows: “If rapid advection from the shallow Holocene aquifer is not responsible for contamination of the upper Pleistocene aquifer with As, the second source of reactive DOC to consider is the thick clay layer itself.”

8. Fig. ED7: Please change units to mM to become consistent with text.

A good suggestion, which was followed.

9. Line 267: Please use 'Because' instead of 'As', not to confuse this 'As' with 'Arsenic'.

Good point - this was done.

Reviewer #3 (Remarks to the Author):

This manuscript on possible in-aquifer exacerbation of arsenic release via clay layers in Bangladesh is thorough, well-written, well-designed and the implications are significant – particularly with regard to the protection, or otherwise, that clay layers above lower As Pleistocene aquifers may give. In general the manuscript seems worthy of publication in Nature Communications. Some points which would benefit from further consideration or clarification:

- It would be helpful to show more characteristics of what this “reactive carbon” source actually is. As far as I can tell, “reactive DOC” here is simply defined as a difference in DOC between unacidified for 1 month and immediately acidified subsamples (around line 941). Is this really a sufficient proxy for “reactive DOC”? This seems a rather big assumption, particularly as it is clear from other studies that not all types of DOC contribute to arsenic release comparably. How do these estimates of “reactive DOC” compare with other bioavailability proxies?

We have an admittedly crude operational indication that the DOC is reactive, but we do not know its exact nature. No one does in fact. Our site might provide a particularly good setting for further study of specific organic compounds and their lability.

- The evidence presented strongly supports that contributions from the in-aquifer clay layer are contributing to arsenic release here. But do we know what exactly from that clay layer is driving this process? The contribution of potential methanogenesis is mentioned (lines 980 – 985) – is there any direct evidence to support this? (e.g. dissolved methane measurements?)

The DOC concentration gradient across the clay/sand interface combined with the inferred reduction of Fe oxides below this interface is the best evidence we can provide. Yes - the upper Pleistocene aquifer contains much higher levels of methane than the deeper orange portion (or the Holocene aquifer above) - and this could be the result of the DOC flux from the clay. This is likely to be a complex story that we are still studying, and therefore bring up only in the Supplementary material - not in the main text. We do not feel that the issue needs to be resolved to make our main point that clay does not necessarily protect and may in fact trigger release of As.

- The heterogeneity of the sub-surface clay layers seems of key importance here, as the authors acknowledge. If drill cuttings of 4 well nests within a 100 m radius indicate a 6 – 13 m thick clay layer, it seems plausible that this clay layer could be very thin, or even missing entirely, in nearby areas too (it is mentioned that the layer is missing 500 m south). Would it be possible that these fast tracks introduced by such windows in the clay layer could contribute surface-derived organics and modern recharge that contribute to the conditions seen here (e.g. tritium increases at 60 mbgl, mixing lines shown on Extended Data 5)? Can this be ruled out?

Such a fast track would likely have been associated with an input of bomb-produced 3H - which is not observed in the grey upper portion of the Pleistocene aquifer. Instead, such flow paths from lateral transport are observed in the deeper orange portion of the aquifer where As concentrations are lower. We now point this out, without claiming actual protection.

Reviewers' comments:

Reviewer #1 (Remarks to the Author):

The authors have robustly and reasonably defended their ms against the comments made by 3 reviewers. They have certainly made some changes in their revised manuscript that have improved the robustness and accuracy of the manuscript however there are a number places where I retain some reservations, notably:

[1] In terms of novelty, the idea of intercalated clay layers providing a critical source of organic carbon with the potential to drive arsenic mobilisation has been published before.

I do agree, though, with the authors, in their rebuttal, that the current ms is novel in that they have uniquely put an outstanding combined geological, geochemical and isotopic database together with the aim of support this idea, particularly in relation to groundwater abstraction driven processes at their study site.

Its clearly an Editorial decision with regard to whether the critical aspect of novelty is related to the idea, the particular driver of a known process or the nature of the dataset, but in my view, it's a very interesting dataset (particularly including the groundwater arsenic time series) in its own right, so I would err on the side of recommending publication, perhaps encouraging the authors to more explicitly state the method-related and driver-related novelty of their work.

[2] The speculative nature of the assertion (albeit tempered down by the use of the word "probably") regarding the orange colour of the Pleistocene sediments at the time of deposition and/or early burial remains. Given the massive importance previously widely attached by many research groups to this orange/grey colour distinction being largely causally associated with low As/high As groundwater respectively, it seems rather odd to claim in the rebuttal that the timing (e.g. thousands of years ago OR last few decades OR during groundwater pumping) of this colour change was not important to arsenic mobilisation in the field area. However, if this IS the case, then the point is rather mute.

[3] My major concern remains the timescales of the observed changes compared to plausible and actual timescales of the arsenic mobilising processes discussed - these concerns are important because they inform what are the major plausible processes leading to the observed changes in groundwater arsenic - particularly noting that the observed changes (Figure 1) are, in various wells and particular time periods, any of (i) increasing; (ii) decreasing and (iii) no trend with time.

In the rebuttal, the authors claim that the increased arsenic in the 41 m-screened well is clearly associated with massive drinking water pumping around Dhaka. That is a very reasonable first-glance hypothesis to test - but the association (let alone any causal link) is NOT clearly demonstrated in this manuscript - not least of all because:

(i) the period of massive pumping in and around Dhaka is not clearly indicated in the manuscript (my apologies if I have missed that);

(ii) the increase in groundwater arsenic in the 41 m well (Figure 1) seems to be largely restricted to the years 2014-2016 - before that the groundwater arsenic time series seems pretty flat, and after 2016 the groundwater arsenic time series also seems pretty flat. Presumably massive groundwater pumping in and around Dhaka was taking place across ALL of these time periods (i.e. 2011-2014, 2014-2016, 2016-2018) ... so its difficult to be convinced of an association between this groundwater pumping and increase in groundwater arsenic from the data presented (particularly in the absence of

the appropriate groundwater abstraction data and any explicit appropriate statistical test/analysis to demonstrate the claimed association)

(iii) over the same period of time, groundwater arsenic in the 51 m well increases between 2011 and 2014 and then there is a pretty flat time series after that; given that groundwater arsenic at 51 m is massively higher than the pre-2014 groundwater arsenic at 41 m - is it not equally or more plausible that the changes observed in groundwater arsenic at the 41 m well rather reflect the movement of higher arsenic groundwaters from elsewhere within the aquifer ? - with this movement accelerated by the increased hydraulic gradients caused by massive groundwater exploitation ? This is obviously a quite different process from that proposed in the manuscript.

(iv) interestingly, as the authors state, in the 64 m well, they observe a DECREASE in groundwater arsenic with time - this could also be readily explained by the same process as in (iii) above - with different directions of groundwater arsenic vs time trends consistent with the well known considerable heterogeneity of aquifer sediments in this region.

It would be useful to have point [3] in particular satisfactorily addressed.

Reviewer #2 (Remarks to the Author):

Review of revised manuscript

Dear Editor, dear authors

In this 2nd round, I first read the authors complete 'rebuttal' document and then I read the manuscript again. Although the authors have only made very tiny changes to the original submission, I must say their text is clearer to me after having read the rebuttal document than before. This I think shows that the authors need to improve their text so that any other first-time-reader will not struggle with concerns, popping up while reading, about the validity of the arguments. By having read the rebuttal I was able to ignore these concerns in my second reading.

To mention an example of the tiny degree of revision: The proposed use of d-excess was agreed with in the rebuttal document, and d-excess has replaced 2H in E.D. Fig. 3. But the main text (lines 189-190) and the Supplementary Discussion still refer to the delta-value of deuterium and not to d-excess. A revision ought to be much more thoroughly carried out than that.

I assume the authors acknowledge that the review process is not mainly intended to be a debate between the authors and the few reviewers. Rather, the reviewers are a 'test-audience' and the authors should welcome any critique and use it to improve their manuscript finally resulting in the best possible published paper.

I would like to point attention to two of my 'major issues' from the 1st round that I think were not satisfactory addressed in the revised manuscript. Perhaps in this second round I am able to be more specific with my concerns. I maintain that I strongly recommend publication in Nat. Comm., but I also

suggest certain improvements prior to final acceptance and publication; with reference to the major issues of the 1st round:

Re. 'Major issue 1':

In the revised manuscript the distinction of deep vs. not-deep(?) Pleistocene aquifer(s) has become more unclear. Now, in the revised ms Pleistocene aquifers are referred to as 'deep' as long as they predate Holocene (<12 ka; first sentence in Abstract). At the same time, the studied Pleistocene aquifer is referred to as 'intermediate' (L46). Lines 42-46 directly suggest that the actually studied Pleistocene aquifer at mainly 40-75 m depth (e.g., Fig. 2) is just a 'more accessible' version of the Pleistocene aquifer at >150 m. Lines 79-80 suggest that we are in fact dealing with a risk for shallow ('upper') Pleistocene aquifers. This confusion is unnecessary. I would say that the Pleistocene aquifer directly underneath Holocene sediments must be referred to as 'shallow Pleistocene', not deep, nor intermediate.

In the rebuttal the authors state that they see no fundamental difference between Pleistocene aquifer materials of different depth. By 'fundamental difference', I assume they refer to the hydrogeology, geochemistry and reactivity related to the potential As release and mobility. To me, an obvious fundamental difference in the context of the study is that the shallow Pleistocene aquifer is capped by (young) Holocene clay (=source of carbon). This is of course not the case for deeper Pleistocene aquifers. Moreover, the studied aquifer sediments are young because they are post-LGM deposits, while the >150 m sediments are much older because they are pre-LGM deposits. For that reason alone I would assume it is very likely that some fundamental differences occur, even if it doesn't show in the variables measured by in the study.

I suggest that the authors revise the manuscript to reflect specifically that their data and conclusions are for a Pleistocene aquifer capped by Holocene clay. Their finding does not need to apply to >150 m deep pleist. aqf. conditions; the conclusions are important enough to show this well for the uppermost Pleistocene aquifer.

Re. 'Major issue 3':

The smoking gun is the current increase in As just below the capping clay, in water without a trace of tritium, etc. as opposed to low As water in modern recharge. This IS the story. But the ms deals a lot with whether or not the grey sediment was oxidized from the start or not. They end up concluding that this actually doesn't matter, but on the way at least two speculative statements are presented which could be deleted if the discussion was organized differently:

Statement 1: 1/3 of the water in the uppermost part of the Pleistocene aquifer stems from the Holocene capping clay.

The statement is based on Cl. However, the delta-value of ^{18}O for the uppermost samples in the aquifer sand (41 m depth) are similar to or perhaps even more negative than in the pore water of the capping Holocene clay and more negative than deeper in the aquifer. These data hence do not support any mixing between the clay and the aquifer, although likely just because no samples of clay pore water were obtained from the very lowest part of the capping clay unit. Accordingly, statement 1 is shown to be wrong by a very good tracer, and therefore the statement is so weakly founded that it should be omitted.

Statement 2: Even though the combined advective and diffusive downwards carbon flux can only bring about 1 meter of sediment from a Fe(II)/Fe ratio of 0.3 to the 0.5 threshold for As release, the flow situation due to the variable elevation of the clay/aquifer interface will create a much deeper reduction into the underlying aquifer.

I disagree. If we assume that the advective leakage from 50 yr pumping + 5000 yr diffusive flux from the bottom of the clay results in a combined carbon flux of 12 mol/m² then vertical dispersive mixing or other types of flow beneath the clay do create more electrons than 12*4 for Fe reduction.

Statement 2 in my view should be deleted.

The authors have the smoking gun. They don't need to explain everything else around it.

Nevertheless: I suggest that the authors try to calculate how much arsenic may be released by the reductive dissolution of 12*4 moles Fe-oxides. I get that the As released will be enough to contaminate lots water to several hundred µg/L:

In one of my previous studies, let's assume that there's just ca. 0.01 µmol As/g sediment (Fig 5, Jessen et al. 2012, GCA) of quite loosely bound As. In 1 m³ of aquifer sand (porosity 0.3) this amounts to 1800 kg/m³ * 1000g/kg * 0.01 µmol/g * 75 µg/µmol = 1,800,000 µg As. If this As is dissolved in the 1 m³'s 300 L of porewater the concentration will be 6000 µg/L. The loosely bound As content is hence enough to contaminate 30 pore volumes of the 1 m³ to 200 µg As/L. Both the high flux of 12 mol C/m² and the many pore volumes would require a high leakage = recent phenomenon caused by Dhaka pumping.

The sediment was probably already reduced from the beginning.

The sudden extra input of carbon is what triggered an addition quick As release which now shows up in the wells.

Best regards
Soren Jessen

Reviewer #3 (Remarks to the Author):

I feel the authors have adequately addressed the review comments raised. Although several of the points are actually still unknown (not only within this study but in the field in general), the authors' statements seem reasonably well-justified and the revisions help to address the concerns raised by all reviewers.

Reviewer #1 (Remarks to the Author):

The authors have robustly and reasonably defended their ms against the comments made by 3 reviewers. They have certainly made some changes in their revised manuscript that have improved the robustness and accuracy of the manuscript however there are a number places where I retain some reservations, notably:

Response: We are glad to see Reviewer #1 recognizes our efforts to revise the manuscript based on previous reviews. This view is somewhat inconsistent with the “very tiny changes” made according to Reviewer # 2 (Dr. Soren Jessen).

[1] In terms of novelty, the idea of intercalated clay layers providing a critical source of organic carbon with the potential to drive arsenic mobilisation has been published before.

I do agree, though, with the authors, in their rebuttal, that the current ms is novel in that they have uniquely put an outstanding combined geological, geochemical and isotopic database together with the aim of support this idea, particularly in relation to groundwater abstraction driven processes at their study site.

Its clearly an Editorial decision with regard to whether the critical aspect of novelty is related to the idea, the particular driver of a known process or the nature of the dataset, but in my view, it's a very interesting dataset (particularly including the groundwater arsenic time series) in its own right, so I would err on the side of recommending publication, perhaps encouraging the authors to more explicitly state the method-related and driver-related novelty of their work.

Response: We are glad to see that Reviewer #1 agrees that our study goes well beyond what has been published on the topic of As release triggered by decompressing clays. In response to the suggestion to be more explicit, we have inserted the following statement in the second to last paragraph of the main text:

“In summary, groundwater As concentrations rose over the past decade in a pre-Holocene aquifer capped by a clay layer. Using multiple lines of evidence, such a rise is attributed here for the first time to the reduction of Fe oxides driven by a flux of reactive carbon originating from a clay layer linked in turn to deep pumping at a considerable distance.”

[2] The speculative nature of the assertion (albeit tempered down by the use of the word "probably") regarding the orange colour of the Pleistocene sediments at the time of deposition and/or early burial remains. Given the massive importance previously widely attached by many research groups to this orange/grey colour distinction being largely causally associated with low As/high As groundwater respectively, it seems rather odd to claim in the rebuttal that the timing (e.g. thousands of years ago OR last few decades OR during groundwater pumping) of this colour change was not important to arsenic mobilisation in the field area. However, if this IS the case, then the point is rather mute.

Response: This study does not challenge the causal link from iron oxide reduction (and color change) to the release of arsenic to groundwater; it builds on it. As Reviewer #1 points out and develops below, the more difficult question to answer definitively is how and when this transformation occurred. When the sand became reduced and changed color to just below the threshold for As release is not a key issue – what forced the system to cross threshold for As release is more relevant here. As explained below, the extended

time series of As monitoring is the best evidence available to address that question – and the interpretation is consistent with other considerations.

[3] My major concern remains the timescales of the observed changes compared to plausible and actual timescales of the arsenic mobilising processes discussed - these concerns are important because they inform what are the major plausible processes leading to the observed changes in groundwater arsenic - particularly noting that the observed changes (Figure 1) are, in various wells and particular time periods, any of (i) increasing; (ii) decreasing and (iii) no trend with time.

In the rebuttal, the authors claim that the increased arsenic in the 41 m-screened well is clearly associated with massive drinking water pumping around Dhaka. That is a very reasonable first-glance hypothesis to test - but the association (let alone any causal link) is NOT clearly demonstrated in this manuscript - not least of all because:

(i) the period of massive pumping in and around Dhaka is not clearly indicated in the manuscript (my apologies if I have missed that);

Response: In the original submission and 1st revision, we only cite Mihajlov et al., WRR 2016 for water elevation time series in a shallow and an intermediate pre-Holocene aquifer at a nearby site extending back to 2001. We agree with the reviewer that a broader temporal perspective was needed and accordingly have inserted an updated version of this time series as a new Extended Data Figure 5, along with additional shorter water elevation series from wells at the study site itself. The water elevation data show that the cone of depression from deep Dhaka pumping started to impact intermediate pre-Holocene aquifers of Araihasar about 20 years ago.

(ii) the increase in groundwater arsenic in the 41 m well (Figure 1) seems to be largely restricted to the years 2014-2016 - before that the groundwater arsenic time series seems pretty flat, and after 2016 the groundwater arsenic time series also seems pretty flat. Presumably massive groundwater pumping in and around Dhaka was taking place across ALL of these time periods (i.e. 2011-2014, 2014-2016, 2016-2018) ... so its difficult to be convinced of an association between this groundwater pumping and increase in groundwater arsenic from the data presented (particularly in the absence of the appropriate groundwater abstraction data and any explicit appropriate statistical test/analysis to demonstrate the claimed association)

Response: We now realize it was not a good idea to display the As time series on a log scale. On a linear scale, the data show roughly parallel trends at 41 m and 51 m. One other point we should have made in the original submission, and we now do in a new section added to the Supplementary Discussion, is that there is no reason to expect a linear response between the degree of iron oxide reduction and As concentrations in groundwater. Data relying on acid-leachable Fe(II) and Fe(II+III) measurements (or the first derivative of the reflectance spectrum as a proxy thereof) from co-occurring sand and groundwater intervals show a threshold of Fe(II)/Fe ~0.5 below which no As is released (see published figures from Araihasar and elsewhere in the Bengal basin below). This means there is no reason either to expect a linear response between the duration of Dhaka pumping and the release of As. The critical Fe(II)/Fe ratio of ~0.5 first needs to be reached, at which point the As (or Fe) released will depend on the degree of reduction but also a host of other factors such as reprecipitation as Fe(II) with some As. Keeping these caveats in mind, linearly extrapolating the As time series at 41 m and 51 m back in time indicates an As content of 5 µg/L representative of un-impacted pre-Holocene aquifers in 2009 and 2003, respectively. This is consistent with

the onset of the divergence in water elevations between the shallow and intermediate aquifer at the study site now shown in the new water elevation figure. We point this out in the 2nd revision without claiming causality solely based on that line of evidence.

[Redacted]

From Horneman et al., GCA 2004

[Redacted]

FIGURE 3. Scatter plots of groundwater and aquifer-solid properties for material collected with the needle sampler in three regions of Bangladesh. Note the logarithmic scale of all properties, with the exception of leachable Fe(II)/Fe. Grey symbols identify 16 out of 45 less reducing intervals characterized by either leachable Fe(II)/Fe < 0.5 or dissolved Fe < 0.2 mg/L and dissolved As < 10 µg/L that were excluded from the regression.

From van Geen et al., EST 2008

(iii) over the same period of time, groundwater arsenic in the 51 m well increases between 2011 and 2014 and then there is a pretty flat time series after that; given that groundwater arsenic at 51 m is massively higher than the pre-2014 groundwater arsenic at 41 m - is it not equally or more plausible that the changes observed in groundwater arsenic at the 41 m well rather reflect the movement of higher arsenic groundwaters from elsewhere within the aquifer ? - with this movement accelerated by the increased hydraulic gradients caused by massive groundwater exploitation ? This is obviously a quite different process from that proposed in the manuscript.

Response: We agree with the reviewer that mixing upward from 51 m to 41 m is an alternative explanation for the increase in As concentration just below the clay worth evaluating. This is also easier to evaluate on a linear concentration scale: a roughly constant 20% contribution from 51 m to 41 m would be consistent with the As data but not the the ³H content of 1.6 TU at 51 m and 0.09 TU at 41 m measured in samples collected in 2011 indicating a 5-10% contribution at most (Extended Data Table 4; no 3H samples

from these wells have been collected since). Even if there had been more mixing, it would only shift the question to the origin of high groundwater As at 51 m. None of the neighboring nests show As levels quite as high in the 40-50 m depth range but the source area could easily have been missed. Alternatively, reduction and As mobilization could have been most advanced at 41 m but already tempered relative to 51 m by reprecipitation of Fe. Higher concentrations of As at 51 m do not necessarily mean that is the direction of the source of DOC. The observations clearly link elevated As in the well at 51 m (which is tapping grey sands and contains ^3H) to processes above the plume of ^3H (such as at the well at 41 m) instead of the processes within the ^3H plume (where no other wells contain elevated As or parameters associated with reduction). The well at 51 m appears to be at a mixing front between the two portions of the aquifer. We have modified the main text of the paper and the Supplementary Material accordingly.

(iv) interestingly, as the authors state, in the 64 m well, they observe a DECREASE in groundwater arsenic with time - this could also be readily explained by the same process as in (iii) above - with different directions of groundwater arsenic vs time trends consistent with the well known considerable heterogeneity of aquifer sediments in this region.

It would be useful to have point [3] in particular satisfactorily addressed.

Response: The magnitude of the decrease in concentrations in the low As well is minimal when considered on a linear scale. The statement was unwarranted and has been deleted.

Reviewer #2 (Remarks to the Author):

Review of revised manuscript

Dear Editor, dear authors

In this 2nd round, I first read the authors complete 'rebuttal' document and then I read the manuscript again. Although the authors have only made very tiny changes to the original submission, I must say their text is clearer to me after having read the rebuttal document than before. This I think shows that the authors need to improve their text so that any other first-time-reader will not struggle with concerns, popping up while reading, about the validity of the arguments. By having read the rebuttal I was able to ignore these concerns in my second reading.

Response: Our understanding is that Nature Communications targets a general scientific audience. Dr. Jessen is one of only a dozen experts in the world who has the ability to think of the caveats to our main argument while reading the paper first time. This is not likely to be the case for most other readers and we have consciously relegated most of the finer points to the Supplemental Material.

To mention an example of the tiny degree of revision: The proposed use of d-excess was agreed with in the rebuttal document, and d-excess has replaced 2H in E.D. Fig. 3. But the main text (lines 189-190) and the Supplementary Discussion still refer to the delta-value of deuterium and not to d-excess. A revision ought to be much more thoroughly carried out than that

Response: This was an oversight and is now corrected – even if we do not think deuterium excess has much to contribute in this particular context.

I assume the authors acknowledge that the review process is not mainly intended to be a debate between the authors and the few reviewers. Rather, the reviewers are a 'test-audience' and the authors should welcome any critique and use it to improve their manuscript finally resulting in the best possible published paper.

Response: We agree entirely and are grateful for Dr. Jessen's time and effort that helped us improve this manuscript.

I would like to point attention to two of my 'major issues' from the 1st round that I think were not satisfactory addressed in the revised manuscript. Perhaps in this second round I am able to be more specific with my concerns. I maintain that I strongly recommend publication in Nat. Comm., but I also suggest certain improvements prior to final acceptance and publication; with reference to the major issues of the 1st round:

Response: We are glad to see that Dr. Jessen continues to recommend publication after his concerns are addressed.

Re. 'Major issue 1':

In the revised manuscript the distinction of deep vs. not-deep(?) Pleistocene aquifer(s) has been come more unclear. Now, in the revised ms Pleistocene aquifers are referred to as 'deep' as long as they predate Holocene (<12 ka; first sentence in Abstract). At the same time, the studied Pleistocene aquifer is referred to as 'intermediate' (L46). Lines 42-46 directly suggest that the actually studied Pleistocene aquifer at mainly 40-75 m depth (e.g., Fig. 2) is just a 'more accessible' version of the Pleistocene aquifer at >150 m. Lines 79-80 suggest that we are in fact dealing with a risk for shallow ('upper') Pleistocene aquifers. This confusion is unnecessary. I would say that the Pleistocene aquifer directly underneath Holocene sediments must be referred to as 'shallow Pleistocene', not deep, nor intermediate.

Response: We appreciate the difficulty in assigning clear names to specific aquifers that are neither continuous or well-defined by their ages or depth. We have chosen a naming convention that is well accepted by the community working in this area, and that is able to distinguish aquifers that may be laterally discontinuous as is common in Bangladesh. While Dr. Jessen suggests a reasonable method of naming aquifers, it is not easy to apply it within the geological setting of Bangladesh. Furthermore, the problem with Dr. Jessen's suggestion is that there are areas in Bangladesh – including the northern portion of our study area in Araihasar – where the shallowest aquifer is of pre-Holocene age and so is most of the capping clay layer (in order to avoid unnecessary geological terms presented to a general audience, we have changed the terminology to Holocene and pre-Holocene rather than Holocene and Pleistocene). "Shallow Pleistocene" would be appropriate for uplifted sites such as in northern Araihasar but not the present study site with an overlying Holocene clay and aquifer. As Dr. Jessen points out in his next comment, the distinction matters as deposition of a more recent Holocene clay is a pre-condition for the mechanism of As release presented in this paper.

In the rebuttal the authors state that they see no fundamental difference between Pleistocene aquifer materials of different depth. By 'fundamental difference', I assume they refer to the hydrogeology, geochemistry and reactivity related to the potential As release and mobility. To me, an obvious fundamental difference in the context of the study is that the shallow Pleistocene aquifer is capped by (young) Holocene clay (=source of carbon). This is of course not the case for deeper Pleistocene aquifers.

Moreover, the studied aquifer sediments are young because they are post-LGM deposits, while the >150 m sediments are much older because they are pre-LGM deposits. For that reason alone I would assume it is very likely that some fundamental differences occur, even if it doesn't shown in the variables measured by in the study.

I suggest that the authors revise the manuscript to reflect specifically that their data and conclusions are for a Pleistocene aquifer capped by Holocene clay. Their finding does not need to apply to >150 m deep pleist. aqf. conditions; the conclusions are important enough to show this well for the uppermost Pleistocene aquifer.

Response: We agree entirely with Dr. Jessen – and refer now to a pre-Holocene aquifer capped by a Holocene clay throughout the paper. We are still not sure this qualifies as a “major issue” – unlike perhaps the next one.

Re. 'Major issue 3':

The smoking gun is the current increase in As just below the capping clay, in water without a trace of tritium, etc. as opposed to low As water in modern recharge. This IS the story. But the ms deals a lot with whether or not the grey sediment was oxidized from the start or not. They end up concluding that this actually doesn't matter, but on the way at least two speculative statements are presented which could be deleted if the discussion was organized differently:

Response: As Reviewer #1 points out, the “smoking gun” referred to by Dr. Jessen is not quite that because the As and ³H data combined cannot rule out that As concentrations in the well at 41 m increase because of upward mixing a ~one-fifth contribution of the water from 51 m (the distance separating these two portions of the aquifer is also closer to 8 m given the 1.5 m screen lengths centered on the two depths). This is explained in the Supplementary Discussion.

Statement 1: 1/3 of the water in the uppermost part of the Pleistocene aquifer the stem from the Holocene capping clay.

The statement is based on Cl. However, the delta-value of 18O for the uppermost samples in the aquifer sand (41 m depth) are similar to or perhaps even more negative than in the pore water of the capping Holocene clay and more negative than deeper in the aquifer. These data hence do not support any mixing between the clay and the aquifer, although likely just because no samples of clay pore water were obtained from the very lowest part of the capping clay unit. Accordingly, statement 1 is shown to be wrong by a very good tracer, and therefore the statement is so weakly founded that it should be omitted.

Response: The Cl-based upper limit for the clay layer contribution is based on two samples squeezed from the very bottom of the clay layer. Not enough pore water could be squeezed for stable isotope measurements – which is why these two samples are missing from the stable water isotope profile. It is therefore not possible to determine if the stable isotope data are consistent with the Cl.

Statement 2: Even though the combined advective and diffusive downwards carbon flux can only bring about 1 meter of sediment from a Fe(II)/Fe ratio of 0.3 to the 0.5 threshold for As release, the flow situation due to the variable elevation of the clay/aquifer interface will create a much deeper reduction into the underlying aquifer.

I disagree. If we assume that the advective leakage from 50 yr pumping + 5000 yr diffusive flux from the bottom of the clay results in a combined carbon flux of 12 mol/m² then vertical dispersive mixing or other types of flow beneath the clay do create more electrons than 12*4 for Fe reduction. Statement 2 in my view should be deleted.

Response: We do not understand how “other types of flow beneath the clay” could increase an electron flux emanating from the capping clay. Dispersion could distribute the electrons over a wider depth range but would not increase the total input and therefore still cannot convert an entire layer of sand from orange to grey. As we point out in our response to Reviewer #1, we think lateral flow from beneath the bottom of different clay layers across the 40-50 depth range is an important element of the overall interpretation.

The authors have the smoking gun. They don't need to explain everything else around it.

Nevertheless: I suggest that the authors try to calculate how much arsenic may be released by the reductive dissolution of 12*4 moles Fe-oxides. I get that the As released will be enough to contaminate lots water to several hundred µg/L:

In one of my previous studies, let's assume that there's just ca. 0.01 µmol As/g sediment (Fig 5, Jessen et al. 2012, GCA) of quite loosely bound As. In 1 m³ of aquifer sand (porosity 0.3) this amounts to 1800 kg/m³ * 1000g/kg * 0.01 µmol/g * 75 µg/µmol = 1,800,000 µg As. If this As is dissolved in the 1 m³'s 300 L of porewater the concentration will be 6000 µg/L. The loosely bound As content is hence enough to contaminate 30 pore volumes of the 1 m³ to 200 µg As/L. Both the high flux of 12 mol C/m² and the many pore volumes would require a high leakage = recent phenomenon caused by Dhaka pumping. *Response: We can all agree that any Bengal basin sands – including pre-Holocene sands – contain more than enough As to increase groundwater concentrations to several thousand µg/L upon reduction (the BGS/DPHE (2001) report was the first to make this important but occasionally overlooked point). We appreciate the value of back-of-the-envelope calculations (and these calculations with respect to As have been included in this and the previous versions – in Supplementary Discussion), but we do not believe this particular one provides a real test of the plausibility of the proposed scenario. The electron flux required to convert orange sand to grey provides a more significant constraint – even if the underlying assumption has its issues too – as Dr. Jessen alludes to next.*

The sediment was probably already reduced from the beginning.

Response: We agree with the back-of-the-envelope calculations that show (a) only a small amount of As in sediments can contaminate a significant volume of water, and (b) the organic carbon needed to liberate it is significant but can only come from a few sources in high enough quantities. Given that the reviewer does not see this clearly from what is written, however, suggests that we need to clarify our writing to emphasize these points and how they related to the generation of high-As groundwater. To do so, we now present the all-orange sand scenario as an upper limit to the flux of DOC needed to reduce Fe oxides to the extent necessary for As release. We point out that the estimated flux of DOC released by diffusion over 5,000 years is insufficient to reduce the entire 10 m layer of sand – unless the variable depth of the bottom of the clay is taken into account. The alternative scenario is that the sand below the clay never was entirely oxidized to the state of the orange layer just below. We think the second scenario is less likely given the continuity in grain-size and Ca content of the entire 40-60 m aquifer but we cannot reject it. Either way – we point out that extrapolating the As time series back in time suggests that Dhaka pumping either

accelerated this release of DOC or, at a minimum, redistributed locally high As groundwater in a way that affected a portion of the aquifer previously low in As. The latter interpretation still requires an explanation for higher As below the clay in the surrounding pre-Holocene aquifer, which we have provided in this paper.

The sudden extra input of carbon is what triggered an additional quick As release which now shows up in the wells.

Best regards

Soren Jessen

Reviewer #3 (Remarks to the Author):

I feel the authors have adequately addressed the review comments raised. Although several of the points are actually still unknown (not only within this study but in the field in general), the authors' statements seem reasonably well-justified and the revisions help to address the concerns raised by all reviewers.

Response: We appreciate the perspective and hope some of the remaining uncertainties were addressed to Reviewer #3's satisfaction in the 2nd revision.

REVIEWERS' COMMENTS:

Reviewer #1 (Remarks to the Author):

Whilst I still do not agree entirely with the conclusions of the manuscript, there is a margin here for disagreement which should not of itself preclude publication.

The authors have substantively revised the manuscript on two occasions in response to reviewers' comments and now have a revised manuscript that, in my opinion, should be published.

No doubt, not all may agree with the conclusions, but the conclusions are now argued reasonably well and will attract interest and debate.

Reviewer #2 (Remarks to the Author):

Comments to 2nd revision of:

Arsenic contamination of Bangladesh aquifers exacerbated by clay layers

Dear Editor, dear Authors

I now find the manuscript very much improved in readability/clarity. The MS is now much more to-the-point and textual precision has been improved in many places.

This time I took the liberty of simply commenting in the MS Word *.docx file. To clarify my points, I further took the liberty of editing directly in the document. I do not mean to push my formulations, but I did this in order to clearly illustrate the (low) level of further modifications to the MS that I suggest the Editor should request before final acceptance.

Reference to line numbers in the following is to the *.docx file, with Microsoft Word's Track Changes set to show 'All Markup'.

As stated, the MS reads very well. But regarding the modifications:

1. I maintain that the authors should clarify the type of aquifer investigated namely aquifers deposited post-LGM and before Holocene, which indeed may contain both grey and orange sands, but which have not had as much time for flushing as a pre-LGM aquifers and also which have not been subject to deep unsaturated oxidation. I acknowledge the author's difficulty in assigning clear names, but I think my modification may solve the problem. In the attached document please see my modifications at lines:

Lines 43-45, 122-123, 127-128, 142, 258-259, 292-293

Also, please read the comments 2, 5, and 8.

I hope the authors will note that these clarifications do not take away the high relevance of their study for the many deep pre-LGM Pleistocene aquifers! Their study is of course high relevant for these – it just was not *directly* what was studied by the authors.

2. The authors should clarify the text at lines 137-142, which currently indicates that the investigated pre-Holocene aquifer above 80 m depth was deposited before the LGM, while their Fig. 2a shows that all sediment samples above 80 m depth post-date the LGM.

Please see comment 6.

3. The authors should rethink their "second explanation" for the triggering of As release. I suggest they simply delete it, or that they provide more evidence for it to be a likely scenario. Please see lines 268-277 and 284-286, and comment no. 9.

4. Minors: I made a number of small modifications/clarifications and typo corrections while running through the text. Please see comment nos. 1, 3, 4, 7 and 10, and lines 41, 46, 47, 52, 53, 64, 81-82, 83, 152, 168-169, 172, 183, 225, 228, 229, 230-231, 254, 265, 461, 767-768, 770, and 794.

Kind regards

Soren